# Intrinsically/Purely Gapless-SPT
# from Non-Invertible Duality Transformations

Linhao Li[1], Masaki Oshikawa[1,2], and Yunqin Zheng[1,2]

[1]  Institute for Solid State Physics,
University of Tokyo, Kashiwa, Chiba 277-8581, Japan
[2]  Kavli Institute for the Physics and Mathematics of the Universe,
University of Tokyo, Kashiwa, Chiba 277-8583, Japan

The Kennedy-Tasaki (KT) transformation was used to construct the gapped symmetry protected topological (SPT) phase from the symmetry breaking phase with open boundary condition, and was generalized in our proceeding work [1] on a ring by sacrificing the unitarity, and should be understood as a non-invertible duality transformation. In this work, we further apply the KT transformation to systematically construct gapless symmetry protected topological phases. This construction reproduces the known examples of (intrinsically) gapless SPT where the non-trivial topological features come from the gapped sectors by means of decorated defect constructions. We also construct new (intrinsically) purely gapless SPTs where there are no gapped sectors, hence are beyond the decorated defect construction. This construction elucidates the field theory description of the various gapless SPTs, and can also be applied to analytically study the stability of various gapless SPT models on the lattice under certain symmetric perturbations.

# 1  Introduction and summary

**Overview of gapless SPT:**  Gapless symmetry protected topological (SPT) systems have received intensive attention recently [2–10], which are a family of gapless systems exhibiting topological features analogous to the gapped SPT phases.

So far, there are a number of constructions of gapless SPT systems, which we briefly review here. The authors of [2], for the first time, constructed the gapless SPT (gSPT) using the decorated defect construction [11, 12], for example, with $G \times H$ global symmetry. The construction applies to any dimensions. The idea is to decorate the $G$ defect of the $G$ symmetric gapless system/CFT (with one vacuum) with a $H$ gapped SPT. The decorated defect construction introduces a gapped sector, under which both $H$ and $G$ act. However, $H$ does not act on the low energy gapless sector. A common feature of the gapless SPT of this kind is that the same topological features (including non-trivial ground state charge under twisted boundary conditions) can also be realized in a gapped $G \times H$ SPT, hence is not "intrinsic" to gapless SPT [13]. It has been shown that this type of gSPT states can exist at the critical points/regime between spontaneously symmetry breaking (SSB) phases and gapped SPT phases. Recent research proposed that gSPT states at criticality between the Haldane phase and various SSB phases can have a possible experimental realization in the lattice systems with bosons [14–19] and fermions [20–23].

In [3], the authors proposed a systematic construction beyond the previous one, where the non-trivial topological features do not have a counter part in the gapped SPT, hence is termed the intrinsically gapless SPT (igSPT). The schematic idea is as follows. The total symmetry is $\Gamma$, fitting into the extension $1 \to H \to \Gamma \to G \to 1$. One starts with a $G$ symmetric gapless system/CFT (with a unique vacuum) with a $G$ self anomaly $\omega_G$. One further stacks an $H$ SPT on top of $G$ defects. However, due to the non-trivial group extension, the induced gapped sector has an opposite $G$ anomaly $-\omega_G$. This cancels the anomaly in the gapless sector, and the combined system is $\Gamma$ anomaly free. The combined system is termed igSPT. By construction, the igSPT also contains a gapped sector coming from the decorated defect construction. Moreover, the topological features can not be realized in a $\Gamma$ gapped SPT, which justifies the name "intrinsic". In [4], we also analyzed concrete analytically tractable spin models of the gSPT and the igSPT in detail, with $\mathbb{Z}_2 \times \mathbb{Z}_2$ symmetry and $\mathbb{Z}_4$ symmetry respectively in $(1+1)$d. These two models will be crucial for the discussions in the main text below. Recently, the authors of [5] discussed a more systematic construction. Besides, the authors of [24] constructed a spin-1 model which hosts both gSPT and igSPT simultaneously. A 2+1 dimensional igSPT example at the deconfined transition between a quantum spin Hall insulator and a superconductor is discussed in more detail in [8].

Both the gSPT and the igSPT contain a gapped sector by construction, which results in the exponential decaying energy splitting of edge modes. A natural question is whether there are gapless SPTs without gapped sectors, hence is purely gapless SPT? In [6], the authors studied one interesting model with time reversal symmetry, and demonstrated that this model does not have a

| | Contains gapped sector | No gapped sector |
|---|---|---|
| Non-intrinsic | gapless SPT [2, 4, 6, 24] | purely gapless SPT [6] |
| Intrinsic | intrinsically gapless SPT [3–5, 8, 24] | intrinsically purely gapless SPT |

Table 1: Classification of gapless SPTs by whether they are purely gapless (horizontal direction) and intrinsically gapless (vertical direction).

gapped sector by noting that the energy splitting under OBC is polynomial. Moreover, the ground state also exhibits non-trivial topological features that admit a gapped counterpart. Following [5] we name it as purely gapless SPT (pgSPT). Moreover, in [5], the authors also put forward an open question about the existence of an intrinsic gapless SPT without gapped sector, i.e. intrinsically purely gapless SPT. In the present work, we answer this question in the affirmative. Finally, one may naturally wonder if there is a more systematic construction of a family of pgSPT and ipgSPT analogue to the decorated defect construction of the gSPT and the igSPT. We confirm this by constructing all the above gapless SPTs using the Kennedy-Tasaki transformation, for simple symmetries. A more systematic exploration with general symmetries will be left to the future.

We summarize the current understanding of gapless SPT in Table 1.

**Kennedy Tasaki transformation:** The Kennedy-Tasaki (KT) transformation was originally found to map the Haldane's spin-1 chain in the $\mathbb{Z}_2 \times \mathbb{Z}_2$ symmetry spontaneously broken (SSB) phase to a "hidden $\mathbb{Z}_2 \times \mathbb{Z}_2$ symmetry breaking phase" [25, 26]. Such a transformation was first found to have a simple compact form

$$U_{\mathrm{KT}} = \prod_{i>j} \exp\left(i\pi S_i^z S_j^x\right) \tag{1.1}$$

by one of the authors of the present work [27]. It is a unitary and highly non-local operator. The "hidden $\mathbb{Z}_2 \times \mathbb{Z}_2$ symmetry breaking phase" is now well-known as the Haldane phase or $\mathbb{Z}_2 \times \mathbb{Z}_2$ gapped SPT phase [28–30]. It is one of the earliest realizations of symmetry protected topological phenomena in condensed matter physics.

The KT transformation (1.1) discussed in [25, 26] was for spin-1 systems with open boundary condition, and how to define it on a ring remains open until recently. In our recent work [1], we proposed a way to realize the KT transformation on a ring, and discussed both spin-1 and spin-$\frac{1}{2}$ chains, which were proven equivalent. We note that, the KT transformation generalized to a spin-$\frac{1}{2}$ in two dimensions with a subsystem symmetry was discussed in [31, App. B], apparently while being unaware of the original KT transformation found in 1990s. Interestingly, the KT transformation on a ring is implemented by the non-unitary operators $\mathcal{N}_{\mathrm{KT}}$ and obeys the non-invertible fusion rule, while it becomes a unitary transformation under a suitable boundary condition on an

interval. We thus name the KT transformation as a non-invertible duality transformation. The effect of KT transformation is to gauge the $\mathbb{Z}_2 \times \mathbb{Z}_2$ with certain topological twists which we specify in the main text. [1]

For convenience, we will focus on spin-$\frac{1}{2}$ systems throughout.

**Constructing (gapped and gapless) SPTs from Kennedy Tasaki transformation:** In this work, we propose to use the KT transformation to construct known examples of gSPT with global symmetry $\mathbb{Z}_2 \times \mathbb{Z}_2$ and igSPT with global symmetry $\mathbb{Z}_4$, and show that our construction automatically gives rise to decorated defect construction in [4] when gapped sectors exist. We also apply the KT transformation to construct possibly the first examples of pgSPT and ipgSPT with only the on-site global symmetries (and not time reversal symmetry). The similar role of the original KT transformation in the gSPT of the spin-1 system is also discussed in the work [24].

We summarize the main results as follows.

$$\mathbb{Z}_2^\sigma \text{ SSB} + \mathbb{Z}_2^\tau \text{ SSB} \quad \overset{\mathcal{N}_{\text{KT}}}{\Longleftrightarrow} \quad \mathbb{Z}_2^\sigma \times \mathbb{Z}_2^\tau \text{ gapped SPT} \tag{1.2}$$

$$\mathbb{Z}_2^\sigma \text{ SSB} + \mathbb{Z}_2^\tau \text{ trivial} \quad \overset{\mathcal{N}_{\text{KT}}}{\Longleftrightarrow} \quad \mathbb{Z}_2^\sigma \text{ SSB} + \mathbb{Z}_2^\tau \text{ trivial} \tag{1.3}$$

$$\mathbb{Z}_2^\sigma \text{ trivial} + \mathbb{Z}_2^\tau \text{ trivial} \quad \overset{\mathcal{N}_{\text{KT}}}{\Longleftrightarrow} \quad \mathbb{Z}_2^\sigma \text{trivial} + \mathbb{Z}_2^\tau \text{trivial} \tag{1.4}$$

$$\mathbb{Z}_2^\sigma \text{ Ising CFT} + \mathbb{Z}_2^\tau \text{ SSB} \quad \overset{\mathcal{N}_{\text{KT}}}{\Longleftrightarrow} \quad \mathbb{Z}_2^\sigma \times \mathbb{Z}_2^\tau \text{ gapless SPT} \tag{1.5}$$

$$\mathbb{Z}_2^\sigma \text{ Ising CFT} + \mathbb{Z}_2^\tau \text{ Ising CFT} \quad \overset{\mathcal{N}_{\text{KT}}}{\Longleftrightarrow} \quad \text{SPT-trivial critical point} \tag{1.6}$$

$$\mathbb{Z}_2^\sigma \text{ SSB} + \mathbb{Z}_4^\tau \text{ free boson CFT} \quad \overset{\mathcal{N}_{\text{KT}}}{\Longleftrightarrow} \quad \mathbb{Z}_4^\Gamma \text{ intrinsically gapless SPT} \tag{1.7}$$

$$\mathbb{Z}_2^\sigma \text{ free boson CFT} + \mathbb{Z}_2^\tau \text{ free boson CFT} \quad \overset{\mathcal{N}_{\text{KT}}}{\Longleftrightarrow} \quad \mathbb{Z}_2^\sigma \times \mathbb{Z}_2^\tau \text{ purely gapless SPT} \tag{1.8}$$

$$\mathbb{Z}_2^\sigma \text{ free boson CFT} + \mathbb{Z}_4^\tau \text{ free boson CFT} \quad \overset{\mathcal{N}_{\text{KT}}}{\Longleftrightarrow} \quad \mathbb{Z}_4^\Gamma \text{ intrinsically purely gapless SPT} \tag{1.9}$$

Eqs. (1.2), (1.3), and (1.4) covering gapped phases of $\mathbb{Z}_2 \times \mathbb{Z}_2$ symmetric systems are already discussed in Ref. [1], but included here for the sake of completeness and illustration. In particular, the $\mathbb{Z}_2^\sigma \times \mathbb{Z}_2^\tau$ gapped SPT can be obtained by starting with decoupled $\mathbb{Z}_2^\sigma$ SSB phase and $\mathbb{Z}_2^\tau$ SSB phase and perform the KT transformation. These mappings between gapped phases were essentially identical to those in the earlier works [26, 39, 40] on the KT transformation, although the present formulation [1] on $S = 1/2$ chain is more convenient for construction of gapless phases.

First, replacing the $\mathbb{Z}_2^\sigma$ SSB phase in the left-hand side of (1.2) with the $\mathbb{Z}_2^\sigma$ Ising CFT, we obtain the $\mathbb{Z}_2^\sigma \times \mathbb{Z}_2^\tau$ gSPT after the mapping (1.6). If we further replace the other $\mathbb{Z}_2^\tau$ SSB phase with the $\mathbb{Z}_2^\sigma$ Ising CFT (1.5), by the KT transformation we obtain the gapless theory corresponding to the critical point between the $\mathbb{Z}_2^\sigma \times \mathbb{Z}_2^\tau$ gapped SPT and trivial phases, as can be seen from (1.2)

---

[1]The topological operator implementing the twisted gauging has been discussed extensively in terms of duality defects. See the recent reviews, e.g. [32, 33], and references therein for more details. We emphasize that the KT transform in the present work maps one theory to another, and is not a symmetry of a single theory. Relatedly, the Kramers-Wannier (KW) duality transformation between two theories and the KW duality operator within one theory have been discussed in [1, 34–37] and [38] respectively.

and (1.4). While this is also gapless, it has an emergent symmetry which has a mixed anomaly with the original symmetry protecting the gapped SPT [41–43] and does not belong to gapless SPT phases we focus on in this paper.

Furthermore, replacing the $\mathbb{Z}_2^\tau$ SSB in (1.2) by $\mathbb{Z}_4^\tau$ symmetric free boson CFT (realized by XX chain on the lattice), we obtain the $\mathbb{Z}_4^\Gamma$ intrinsically gapless SPT where $\mathbb{Z}_4^\Gamma$ is generated by the product of generators of $\mathbb{Z}_2^\sigma$ and $\mathbb{Z}_4^\tau$. Replacing both $\mathbb{Z}_2^\sigma$ and $\mathbb{Z}_2^\tau$ SSB by $\mathbb{Z}_2^\sigma$ and $\mathbb{Z}_2^\tau$ free boson CFTs respectively, we obtain the $\mathbb{Z}_2^\sigma \times \mathbb{Z}_2^\tau$ pgSPT. Finally replacing $\mathbb{Z}_2^\sigma$ SSB in (1.2) by $\mathbb{Z}_2^\sigma$ free boson CFT, and $\mathbb{Z}_2^\tau$ SSB in (1.2) by $\mathbb{Z}_4^\tau$ free boson CFT, we obtain $\mathbb{Z}_4^\Gamma$ ipgSPT.

**Advantages of the present construction of gapless SPT phases using the KT transformation:** The common feature of the above constructions is that we start with a decoupled system, which can be either gapped or gapless, and KT transformation will map it to a coupled system with interesting topological features. This construction enables us to construct not only the known models of gSPT and igSPT, but also the new models of pgSPT and ipgSPT. Furthermore, it allows us to study the stability of various gapless SPTs from (1.5) to (1.9) under certain symmetric perturbations. In particular, if the perturbation of the gapless SPT is such that by undoing KT transformation the theory is still decoupled, we can analytically investigate the topological features of the decoupled theory on both a ring and an interval, and then use the KT transformation to trace these topological features back to gapless SPTs of interest. Indeed, this leads to an analytical understanding of the phase diagram of a nontrivial model, which we were only able to investigate numerically in [4].

Gapless SPT phases are often characterized by edge states, which appear as low-energy states in the energy spectrum of an open chain and can be distinguished from low-energy gapless excitations in the bulk. Although such distinction between the gapless excitations and the edge states are possible, it is more subtle compared to identification of edge states in gapped SPT phases. By the KT transformation, we can often relate the low-energy states due to the edge states to the quasi-degeneracy of finite-size ground states due to spontaneous symmetry breaking. This clarifies the identification of the edge states in gapless SPTs and their stability, which also underscores the analysis of the phase diagram as discussed above.

Finally, we also remark that, as the theories on the left hand side of (1.5) to (1.9) admit field theory descriptions, we are able to derive the field theory description of the gapless SPTs of interest, using the KT transformation.

**Outlook and the structure of the present paper:** Although we will only study the gapless SPTs with simple global symmetries like $\mathbb{Z}_2 \times \mathbb{Z}_2$ or $\mathbb{Z}_4$, the KT transformation and the construction of these interesting gapless topological systems can be straightforwardly generalized to more general symmetry groups, as well as to higher dimensions [37]. It is also interesting to explore what condition we should impose on the two decoupled theories so that under (suitably generalized) KT transformation they give rise to gapless SPTs. We will leave these questions to future studies.

The plan of this paper is as follows. In Section 2, we review the basic properties of KT transformation from [1]. In Section 3, we revisit how to use the KT transformation to construct $\mathbb{Z}_2^\sigma \times \mathbb{Z}_2^\tau$ gapped SPT starting from two decoupled $\mathbb{Z}_2$ SSB systems. In Section 4, Section 5 and Section 6, we construct the gapless SPT, intrinsically gapless SPT and purely gapless SPT as well as intrinsically purely gapless SPT respectively. We discuss how to use KT transformation to construct these models, how to probe the topological features, and how to analytically study the phase diagrams under certain symmetric perturbations.

# 2   Review of Kennedy-Tasaki transformation

In this section, we review the Kennedy-Tasaki (KT) transformation defined in [1], which is well-defined under both closed and open boundary conditions. By definition, this new KT transformation is defined by implementing $STS$ on a $\mathbb{Z}_2 \times \mathbb{Z}_2$ symmetric system, where both $\mathbb{Z}_2$'s are anomaly free. Here $S$ is gauging of both $\mathbb{Z}_2$'s, and $T$ is stacking the system with a $\mathbb{Z}_2 \times \mathbb{Z}_2$ bosonic gapped SPT phase.

## 2.1   Definition of the KT transformation for spin-$\frac{1}{2}$ chains

Let us consider a spin chain with $L$ sites and $L$ links. Each site supports one spin-$\frac{1}{2}$, spanning a two dimensional local Hilbert space $|s_i^\sigma\rangle$, where $s_i^\sigma = 0, 1$ and $i = 1, ..., L$. Moreover, each link also supports one spin-$\frac{1}{2}$ spanning a two dimensional local Hilbert space $|s_{i-\frac{1}{2}}^\tau\rangle$, where $s_{i-\frac{1}{2}}^\tau = 0, 1$ for $i = 1, ..., L$. Hence each unit cell contains two spin-$\frac{1}{2}$'s. The local states can be acted upon by Pauli operators,

$$\sigma_i^z |s_i^\sigma\rangle = (-1)^{s_i^\sigma} |s_i^\sigma\rangle, \qquad \sigma_i^x |s_i^\sigma\rangle = |1 - s_i^\sigma\rangle$$
$$\tau_{i-\frac{1}{2}}^z |s_{i-\frac{1}{2}}^\tau\rangle = (-1)^{s_{i-\frac{1}{2}}^\tau} |s_{i-\frac{1}{2}}^\tau\rangle, \qquad \tau_{i-\frac{1}{2}}^x |s_{i-\frac{1}{2}}^\tau\rangle = |1 - s_{i-\frac{1}{2}}^\tau\rangle. \tag{2.1}$$

The $\mathbb{Z}_2 \times \mathbb{Z}_2$ symmetry is generated by $U_\sigma$ and $U_\tau$ respectively, where

$$U_\sigma = \prod_{i=1}^L \sigma_i^x, \qquad U_\tau = \prod_{i=1}^L \tau_{i-\frac{1}{2}}^x. \tag{2.2}$$

The symmetry and twist sectors are labeled by $(u_\sigma, u_\tau, t_\sigma, t_\tau)$. Here $(-1)^{u_\sigma}, (-1)^{u_\tau}$ are the eigenvalues of $U_\sigma, U_\tau$ respectively, and $t_\sigma, t_\tau$ label the boundary conditions $s_{i+L}^\sigma = s_i^\sigma + t_\sigma, s_{i-\frac{1}{2}+L}^\tau = s_{i-\frac{1}{2}}^\tau + t_\tau$. The KT transformation is then defined by the following action on the Hilbert space

basis state [1]

$$\mathcal{N}_{\text{KT}} \left| \{s_i^\sigma, s_{i-\frac{1}{2}}^\tau\} \right\rangle = \frac{1}{2^{L-1}} \sum_{\{s_i'^\sigma, s_{i-\frac{1}{2}}'^\tau\}} (-1)^{\sum_{j=1}^L (s_j^\sigma + s_j'^\sigma)(s_{j-\frac{1}{2}}^\tau + s_{j+\frac{1}{2}}^\tau + s_{j-\frac{1}{2}}'^\tau + s_{j+\frac{1}{2}}'^\tau) + (s_{\frac{1}{2}}^\tau + s_{\frac{1}{2}}'^\tau)(t_\sigma + t_\sigma')} \left| \{s_i'^\sigma, s_{i-\frac{1}{2}}'^\tau\} \right\rangle$$

$$= \frac{1}{2^{L-1}} \sum_{\{s_i'^\sigma, s_{i-\frac{1}{2}}'^\tau\}} (-1)^{\sum_{j=1}^L (s_{j-\frac{1}{2}}^\tau + s_{j-\frac{1}{2}}'^\tau)(s_{j-1}^\sigma + s_j^\sigma + s_{j-1}'^\sigma + s_j'^\sigma) + (s_L^\sigma + s_L'^\sigma)(t_\tau + t_\tau')} \left| \{s_i'^\sigma, s_{i-\frac{1}{2}}'^\tau\} \right\rangle$$

(2.3)

where we have presented two equivalent expressions, which will be convenient for the applications later.

It is useful to emphasize that the original KT transformation is defined for spin-1 systems, while this KT transformation is defined for a spin chain with two spin-$\frac{1}{2}$ per unit cell. Although in [1] we have shown that they are equivalent, it turns out to be more convenient to use the latter set up for the entire discussions, which we will assume throughout this work.

## 2.2   Properties of KT trnasformation

In [1], various properties of (2.3) are examined, including the mapping between symmetry and twist sectors, the fusion rule of the non-invertible defects, the definition on open boundary conditions and the relation to the original KT transformations in the spin-1 models [25–27]. We briefly review the results and refer interested readers to [1] for details.

**Mapping between symmetry-twist sectors:**   Suppose a state is within the symmetry-twist sector labeled by $[(u_\sigma, t_\sigma), (u_\tau, t_\tau)]$, then under the KT transformation, the resulting state is within the symmetry-twist sector labeled by

$$[(u_\sigma', t_\sigma'), (u_\tau', t_\tau')] = [(u_\sigma, t_\sigma + u_\tau), (u_\tau, t_\tau + u_\sigma)]. \tag{2.4}$$

In the sections below, we will frequently use the following result. Suppose the Hamiltonian $H'$ is obtained from $H$ by a KT transformation, i.e. $H'\mathcal{N}_{\text{KT}} = \mathcal{N}_{\text{KT}}H$. If $|\psi_{[(u_\sigma, t_\sigma), (u_\tau, t_\tau)]}\rangle$ is an eigenstate of $H$ in the symmetry-twist sector $[(u_\sigma, t_\sigma), (u_\tau, t_\tau)]$ with energy $E^H_{[(u_\sigma, t_\sigma), (u_\tau, t_\tau)]}$, then $\mathcal{N}_{\text{KT}} |\psi_{[(u_\sigma, t_\sigma), (u_\tau, t_\tau)]}\rangle$ is an eigenstate of $H'$ in the symmetry-twist sector $|\psi_{[(u_\sigma, t_\sigma), (u_\tau, t_\tau)]}\rangle$ with the same energy $E^H_{[(u_\sigma, t_\sigma), (u_\tau, t_\tau)]}$:

$$H'\mathcal{N}_{\text{KT}} |\psi_{[(u_\sigma, t_\sigma), (u_\tau, t_\tau)]}\rangle = \mathcal{N}_{\text{KT}}H |\psi_{[(u_\sigma, t_\sigma), (u_\tau, t_\tau)]}\rangle = E^H_{[(u_\sigma, t_\sigma), (u_\tau, t_\tau)]}\mathcal{N}_{\text{KT}} |\psi_{[(u_\sigma, t_\sigma), (u_\tau, t_\tau)]}\rangle.$$

(2.5)

Note that $\mathcal{N}_{\text{KT}} |\psi_{[(u_\sigma, t_\sigma), (u_\tau, t_\tau)]}\rangle$ sits in the symmetry-twist sector $[(u_\sigma', t_\sigma'), (u_\tau', t_\tau')]$, hence

$$E^{H'}_{[(u_\sigma', t_\sigma'), (u_\tau', t_\tau')]} = E^H_{[(u_\sigma, t_\sigma), (u_\tau, t_\tau)]} = E^H_{[(u_\sigma', t_\sigma' + u_\tau'), (u_\tau', t_\tau' + u_\sigma')]}. \tag{2.6}$$

We will use (2.6) repeatedly in the subsequent sections.

**Fusion rules:** The fusion rules involving the operator implementing the KT transformation, $\mathcal{N}_{\mathrm{KT}}$, and the $\mathbb{Z}_2 \times \mathbb{Z}_2$ symmetry operators $U_\sigma, U_\tau$ are

$$\mathcal{N}_{\mathrm{KT}} \times U_\sigma = (-1)^{t_\tau + t'_\tau} \mathcal{N}_{\mathrm{KT}},$$

$$\mathcal{N}_{\mathrm{KT}} \times U_\tau = (-1)^{t_\sigma + t'_\sigma} \mathcal{N}_{\mathrm{KT}}, \tag{2.7}$$

$$\mathcal{N}_{\mathrm{KT}} \times \mathcal{N}_{\mathrm{KT}} = 4(1 + (-1)^{t_\sigma + t'_\sigma} U_\tau)(1 + (-1)^{t_\tau + t'_\tau} U_\sigma).$$

In particular, the last fusion rule shows that $\mathcal{N}_{\mathrm{KT}}$ is non-invertible, and the transformation is non-unitary.

All the above discussions are on a ring. We finally note that on an open interval, the KT transformation is a unitary transformation, hence preserves the energy eigenvalues of the Hamiltonian [1].

# 3   Gapped SPT from KT transformation

The KT transformation was designed to map a $\mathbb{Z}_2 \times \mathbb{Z}_2$ symmetry spontaneously broken (SSB) phase to a $\mathbb{Z}_2 \times \mathbb{Z}_2$ symmetry protected topological (SPT) phase. It is straightforward to check at the level of partition function that $STS$ transformation relates the two. We will review how the SPT phase can be generated from the KT transformation, and this will be the first example of using the KT transformation to generate exotic models with interesting topological features.

**Gapped SPT from KT transformation:** The Hamiltonian for the $\mathbb{Z}_2 \times \mathbb{Z}_2$ SSB phase is

$$H_{\mathrm{SSB}} = -\sum_{i=1}^{L} \left( \sigma_{i-1}^z \sigma_i^z + \tau_{i-\frac{1}{2}}^z \tau_{i+\frac{1}{2}}^z \right) \tag{3.1}$$

where the degrees of freedom charged under two $\mathbb{Z}_2$'s are decoupled. Under the KT transformation, the operators are mapped as follows

$$\mathcal{N}_{\mathrm{KT}} \sigma_{j-1}^z \sigma_j^z = \sigma_{j-1}^z \tau_{j-\frac{1}{2}}^x \sigma_j^z \mathcal{N}_{\mathrm{KT}}, \qquad \mathcal{N}_{\mathrm{KT}} \tau_{i-\frac{1}{2}}^z \tau_{i+\frac{1}{2}}^z = \tau_{j-\frac{1}{2}}^z \sigma_j^x \tau_{j+\frac{1}{2}}^z \mathcal{N}_{\mathrm{KT}} \tag{3.2}$$

for $j = 1, ..., L$. Note that the boundary condition is encoded in the states/operators already. For instance, the boundary condition $s_L^\sigma = s_0^\sigma + t_\sigma$ induces $\sigma_0^z = (-1)^{t_\sigma} \sigma_L^z$. The resulting Hamiltonian is precisely the cluster model describing the $\mathbb{Z}_2 \times \mathbb{Z}_2$ gapped SPT [44]

$$H_{\mathrm{SPT}} = -\sum_{j=1}^{L} \left( \sigma_{j-1}^z \tau_{j-\frac{1}{2}}^x \sigma_j^z + \tau_{j-\frac{1}{2}}^z \sigma_j^x \tau_{j+\frac{1}{2}}^z \right). \tag{3.3}$$

Let's also comment on the field theory of the gapped SPT. We start with the $\mathbb{Z}_2^\sigma \times \mathbb{Z}_2^\tau$ SSB phase, whose partition function is $Z_{\mathrm{SSB}}[A_\sigma, A_\tau] := \delta(A_\sigma) \delta(A_\tau)$. We define the topological manipulations $S$ and $T$ on a generic quantum field theory $\mathcal{X}$ with $\mathbb{Z}_2^\sigma \times \mathbb{Z}_2^\tau$ symmetry as

$$S: \quad Z_{S\mathcal{X}}[A_\sigma, A_\tau] = \sum_{a_\sigma, a_\tau} Z_{\mathcal{X}}[a_\sigma, a_\tau] e^{i\pi \int_{X_2} a_\sigma A_\tau - a_\tau A_\sigma},$$

$$T: \quad Z_{T\mathcal{X}}[A_\sigma, A_\tau] = Z_{\mathcal{X}}[A_\sigma, A_\tau] e^{i\pi \int_{X_2} A_\sigma A_\tau}. \tag{3.4}$$

In [1], it was found that the KT transformation is $STS$, under which the SSB partition function $Z_{\text{SSB}}[A_\sigma, A_\tau]$ is mapped to

$$Z_{\text{SPT}}[A_\sigma, A_\tau] = \sum_{a_\sigma, a_\tau} \delta(a_\sigma) \delta(a_\tau) e^{i\pi \int_{X_2} a_\sigma \tilde{a}_\tau + a_\tau \tilde{a}_\sigma + \tilde{a}_\sigma \tilde{a}_\tau + \tilde{a}_\sigma A_\tau + \tilde{a}_\tau A_\sigma} = e^{i\pi \int_{X_2} A_\sigma A_\tau} \tag{3.5}$$

which is merely an invertible phase in terms of the background fields. This is commonly known as the field theory description of the gapped SPT [45]. [2]

**Ground state charge under TBC:** A key feature of the SPT is that the ground state carries a non-trivial charge under twisted boundary conditions. To see this, we note that every two terms in (3.3) commute, hence the ground state should be a common eigenstate of each local operator in (3.3). Under PBC for both $\mathbb{Z}_2$'s, the ground state satisfies

$$\sigma_{i-1}^z \tau_{i-\frac{1}{2}}^x \sigma_i^z |\psi\rangle_{\text{PBC}} = |\psi\rangle_{\text{PBC}}, \qquad \tau_{i-\frac{1}{2}}^z \sigma_i^x \tau_{i+\frac{1}{2}}^z |\psi\rangle_{\text{PBC}} = |\psi\rangle_{\text{PBC}} \tag{3.6}$$

for $i = 1, ..., L$, where $\tau_{L+\frac{1}{2}}^z = \tau_{\frac{1}{2}}^z$, and $\sigma_0^z = \sigma_L^z$. Hence

$$U_\sigma |\psi\rangle_{\text{PBC}} = \prod_{i=1}^{L} \sigma_i^x |\psi\rangle_{\text{PBC}} = \prod_{i=1}^{L} \tau_{i-\frac{1}{2}}^z \tau_{i+\frac{1}{2}}^z |\psi\rangle_{\text{PBC}} = |\psi\rangle_{\text{PBC}},$$

$$U_\tau |\psi\rangle_{\text{PBC}} = \prod_{i=1}^{L} \tau_{i-\frac{1}{2}}^x |\psi\rangle_{\text{PBC}} = \prod_{i=1}^{L} \sigma_{i-1}^z \sigma_i^z |\psi\rangle_{\text{PBC}} = |\psi\rangle_{\text{PBC}} \tag{3.7}$$

where we have used $\sigma_0^z = \sigma_L^z$ and $\tau_{\frac{1}{2}}^z = \tau_{L+\frac{1}{2}}^z$ for PBC. Hence the ground state under PBC is neutral under $\mathbb{Z}_2^\sigma \times \mathbb{Z}_2^\tau$.

Under TBC of $\mathbb{Z}_2^\sigma$, if writing the Hamiltonian in terms of the Pauli operators supported within $i = 1, ..., L$, the sign of the term $\sigma_0^z \tau_{\frac{1}{2}}^x \sigma_1^z = -\sigma_L^z \tau_{\frac{1}{2}}^x \sigma_1^z$ changes sign, and the ground state in the TBC of $\mathbb{Z}_2^\sigma$ satisfies

$$\sigma_{i-1}^z \tau_{i-\frac{1}{2}}^x \sigma_i^z |\psi\rangle_{\text{TBC}_\sigma} = |\psi\rangle_{\text{TBC}_\sigma}, i = 2, ..., L, \qquad \sigma_L^z \tau_{\frac{1}{2}}^x \sigma_1^z |\psi\rangle_{\text{TBC}_\sigma} = -|\psi\rangle_{\text{TBC}_\sigma},$$

$$\tau_{i-\frac{1}{2}}^z \sigma_i^x \tau_{i+\frac{1}{2}}^z |\psi\rangle_{\text{TBC}_\sigma} = |\psi\rangle_{\text{TBC}_\sigma}, i = 1, ..., L-1, \qquad \tau_{L-\frac{1}{2}}^z \sigma_L^x \tau_{\frac{1}{2}}^z |\psi\rangle_{\text{TBC}_\sigma} = |\psi\rangle_{\text{TBC}_\sigma}. \tag{3.8}$$

Hence

$$U_\sigma |\psi\rangle_{\text{TBC}_\tau} = \prod_{i=1}^{L} \sigma_i^x |\psi\rangle_{\text{TBC}_\tau} = \prod_{i=1}^{L} \tau_{i-\frac{1}{2}}^z \tau_{i+\frac{1}{2}}^z |\psi\rangle_{\text{TBC}_\tau} = |\psi\rangle_{\text{TBC}_\tau},$$

$$U_\tau |\psi\rangle_{\text{TBC}_\sigma} = \prod_{i=1}^{L} \tau_{i-\frac{1}{2}}^x |\psi\rangle_{\text{TBC}_\sigma} = \prod_{i=1}^{L} \sigma_{i-1}^z \sigma_i^z |\psi\rangle_{\text{TBC}_\sigma} = -|\psi\rangle_{\text{TBC}_\sigma} \tag{3.9}$$

---

[2]The construction of gapped SPT from $STS$ is somewhat round about, since $T$ itself is stacking a gapped SPT (or equivalently domain wall decoration). However, this construction will be more useful when constructing gapless SPT in later sections.

where we have used $\sigma_0^z = -\sigma_L^z$, and $\tau_{\frac{1}{2}}^z = \tau_{L+\frac{1}{2}}^z$ for TBC of $\mathbb{Z}_2^\sigma$. Hence the ground state under TBC of $\mathbb{Z}_2^\sigma$ is $\mathbb{Z}_2^\sigma$ even and $\mathbb{Z}_2^\tau$ odd.

This is a key feature of the gapped SPT. Similarly, one can also show that the ground state under the TBC of $\mathbb{Z}_2^\tau$ is $\mathbb{Z}_2^\sigma$ odd and $\mathbb{Z}_2^\tau$ even.

It is useful to see how the topological features of (3.3) discussed above can be uncovered from the KT transformation without solving (3.3). We begin by analyzing the ground states of the SSB phase (3.1) under various boundary conditions. Because (3.1) is a classical model, its energy spectrum is straightforward to find. Concretely, we have

$$E_{(u,t)}^\sigma = E_{(u,t)}^\tau = -L + 2[t]_2 = \begin{cases} -L, & (u,t) = (0,0),(1,0), \\ -L+2, & (u,t) = (0,1),(1,1) \end{cases} \tag{3.10}$$

where $E_{(u,t)}^\sigma$ is the ground state energy of the sigma spin in the symmetry-twist sector $(u,t)$, and $[t]_2$ is the mod 2 value of $t$. Similar for $E_{(u,t)}^\tau$. Then the ground state energy of the SPT Hamiltonian (3.3) in the symmetry-twist sector $[(u_\sigma, t_\sigma),(u_\tau, t_\tau)]$ is

$$E_{[(u_\sigma,t_\sigma),(u_\tau,t_\tau)]}^{\text{SPT}} = E_{(u_\sigma,t_\sigma+u_\tau)}^\sigma + E_{(u_\tau,t_\tau+u_\sigma)}^\tau = -2L + 2[t_\sigma + u_\tau]_2 + 2[t_\tau + u_\sigma]_2 \tag{3.11}$$

where we have used (2.6) in the first equality, and (3.10) in the second equality. The energy (3.11) is minimized if

$$t_\sigma = u_\tau, \qquad t_\tau = u_\sigma. \tag{3.12}$$

This means that the ground state in the $\mathbb{Z}_2^\sigma$ twisted sector carries non-trivial $\mathbb{Z}_2^\tau$ charge, and the ground state in the $\mathbb{Z}_2^\tau$ twisted sector carries non-trivial $\mathbb{Z}_2^\sigma$ charge. This reproduces the key topological features of the gapped SPT phase reviewed above.

We would like to emphasize the power of the latter method. Typically, the symmetry properties of the system before KT transformation are much easier to analyze, and by (2.6) we automatically know the symmetry properties of the system after KT transformation. Below, we will encounter systems which are difficult to analyze after KT transformation, hence the latter method becomes much more powerful.

**String order parameter:** The string order parameters follow from the local order parameters under KT transformation. In the SSB phase, the long range order is given by the conventional correlation functions

$$\langle \sigma_i^z \sigma_j^z \rangle, \qquad \langle \tau_{i-\frac{1}{2}}^z \tau_{j-\frac{1}{2}}^z \rangle, \qquad i < j, \tag{3.13}$$

both of which are of order 1 in the vacuum. Under KT transformation, $\sigma_i^z \sigma_{i+1}^z$ is mapped to $\sigma_i^z \tau_{i+\frac{1}{2}}^x \sigma_{i+1}^z$, and $\tau_{i-\frac{1}{2}}^z \tau_{j-\frac{1}{2}}^z$ is mapped to $\tau_{i-\frac{1}{2}}^z \sigma_i^x \tau_{i+\frac{1}{2}}^z$, thus the above two conventional correlation functions become string order parameters

$$\langle \sigma_i^z (\prod_{k=i}^{j-1} \tau_{k+\frac{1}{2}}^x) \sigma_j^z \rangle, \qquad \langle \tau_{i-\frac{1}{2}}^z (\prod_{k=i}^{j-1} \sigma_k^x) \tau_{j-\frac{1}{2}}^z \rangle. \tag{3.14}$$

Both the string order parameters develop a non-trivial order 1 vacuum expectation value (VEV) in the ground state of SPT.

# 4 Gapless SPT from KT transformation

In Section 3, we constructed the gapped SPT phase from two decoupled copies of $\mathbb{Z}_2$ symmetry breaking phases. Recent years have witnessed extensive studies of gapless SPT states [2–4, 6]. It is then natural to consider whether such states admit constructions from the KT transformation. In this section, we will confirm this possibility and show that the gapless SPT state first found in [2] (see also [4]) can be constructed in this way.

## 4.1 Constructing the gapless SPT

Instead of starting with two decoupled $\mathbb{Z}_2$ SSB models, we start with a $\mathbb{Z}_2$ gapless model (i.e. the transverse field Ising model) and a decoupled $\mathbb{Z}_2$ SSB model. The Hamiltonian is

$$H_{\text{Ising+SSB}} = -\sum_{i=1}^{L} \left( \tau^z_{i-\frac{1}{2}} \tau^z_{i+\frac{1}{2}} + \sigma^z_{i-1} \sigma^z_i + \sigma^x_i \right). \tag{4.1}$$

The $\mathbb{Z}_2^\tau$ is SSB, and the degrees of freedom charged under $\mathbb{Z}_2^\sigma$ are gapless. As usual, we encode the boundary condition in the Hilbert space, and the Hamiltonian applies to arbitrary boundary conditions. To apply the KT transformation, we still use the operator maps (3.2) and also the map

$$\mathcal{N}_{\text{KT}} \sigma^x_i = \sigma^x_i \mathcal{N}_{\text{KT}} \tag{4.2}$$

for $i = 1, ..., L$. The resulting Hamiltonian is

$$H_{\text{gSPT}} = -\sum_{j=1}^{L} \left( \tau^z_{i-\frac{1}{2}} \sigma^x_i \tau^z_{i+\frac{1}{2}} + \sigma^z_{i-1} \tau^x_{i-\frac{1}{2}} \sigma^z_i + \sigma^x_i \right). \tag{4.3}$$

Note that the first term commutes with the last two terms, hence the ground state $|\psi\rangle$ should satisfy $\tau^z_{i-\frac{1}{2}} \sigma^x_i \tau^z_{i+\frac{1}{2}} |\psi\rangle = |\psi\rangle$. This condition is actually also satisfied by the low excited states as well, because violating it would cost energy of order 1, while the excitation gap is only of order $1/L$. See [4] for more detailed discussions on this point. Hence within the low energy sector, $\sigma^x_i$ can be safely replaced by $\tau^z_{i-\frac{1}{2}} \tau^z_{i+\frac{1}{2}}$, and the Hamiltonian (4.3) is equivalent to

$$H_{\text{gSPT}} \simeq -\sum_{j=1}^{L} \left( \tau^z_{i-\frac{1}{2}} \sigma^x_i \tau^z_{i+\frac{1}{2}} + \sigma^z_{i-1} \tau^x_{i-\frac{1}{2}} \sigma^z_i + \tau^z_{i-\frac{1}{2}} \tau^z_{i+\frac{1}{2}} \right). \tag{4.4}$$

This is exactly the Hamiltonian for the gapless SPT originally constructed in [2] and later revisited in [4]. [3] (4.3) and (4.4) are also related by Kramers-Wannier (KW) transformation for both $\mathbb{Z}_2^\sigma \times \mathbb{Z}_2^\tau$.

---

[3]In [2] and [4], the role of $\sigma$ and $\tau$ are exchanged.

## 4.2 Field theory of gapless SPT

The KT transformation also allows us to write down the field theory for the gapless SPT. We start with the partition function for the Ising CFT + SSB phase,

$$Z_{\text{Ising}}[A_\sigma]Z_{\text{SSB}}[A_\tau] \tag{4.5}$$

where the partition function of the Ising CFT can be conveniently written as a Wilson-Fisher fixed point,

$$Z_{\text{Ising}}[A_\sigma] := \int \mathcal{D}\phi \exp\left(i\int_{X_2} (D_{A_\sigma}\phi)^2 + \phi^4\right), \qquad D_{A_\sigma}\phi = d\phi - i\pi A_\sigma \phi \tag{4.6}$$

and the partition function of the $\mathbb{Z}_2^\tau$ SSB phase is simply a delta function restricting its background field to zero,

$$Z_{\text{SSB}}[A_\tau] = \delta(A_\tau). \tag{4.7}$$

We then perform KT transformation, i.e. a $STS$ transformation, changing the partition function to

$$\begin{aligned}
Z_{\text{gSPT}}[A_\sigma, A_\tau] &= \sum_{a_\sigma, a_\tau, \tilde{a}_\sigma, \tilde{a}_\tau} Z_{\text{Ising}}[a_\sigma]Z_{\text{SSB}}[a_\tau]e^{i\pi\int_{X_2} a_\sigma\tilde{a}_\tau + a_\tau\tilde{a}_\sigma + \tilde{a}_\sigma\tilde{a}_\tau + \tilde{a}_\sigma A_\tau + \tilde{a}_\tau A_\sigma} \\
&= \sum_{a_\sigma, a_\tau} Z_{\text{Ising}}[a_\sigma]\delta(a_\tau)e^{i\pi\int_{X_2}(a_\sigma + A_\sigma)(a_\tau + A_\tau)} = \sum_{a_\sigma} Z_{\text{Ising}}[a_\sigma]e^{i\pi\int_{X_2} a_\sigma A_\tau + A_\sigma A_\tau} \\
&\longleftrightarrow Z_{\text{Ising}}[A_\sigma]e^{i\pi\int_{X_2} A_\sigma A_\tau}.
\end{aligned} \tag{4.8}$$

In the last line, we used the Kramers-Wannier duality which identifies the gauged Ising CFT with the Ising CFT itself. Comparing the head and tail of (4.8) shows that the $\mathbb{Z}_2^\sigma \times \mathbb{Z}_2^\tau$ gapless SPT is simply an Ising CFT stacked with a $\mathbb{Z}_2^\sigma \times \mathbb{Z}_2^\tau$ gapped SPT, which matches the construction in [2,4].

## 4.3 Topological features of gapless SPT

We proceed to study the topological features of the gapless SPT directly from the Hamiltonian (4.3). We will focus on the symmetry charges of the ground state under the TBCs, as well as the degeneracy under the open boundary condition. The discussion here follows [4].

### 4.3.1 Symmetry charge of ground state under TBC

For definiteness, we will consider the Hamiltonian (4.3), although (4.4) is equivalent. The discussion is similar to that for the gapped SPT in Section 3. Under PBC, since $\tau^z_{i-\frac{1}{2}}\sigma^x_i\tau^z_{i+\frac{1}{2}}$ commutes with the remaining terms, the ground state $|\psi\rangle_{\text{PBC}}$ must be an eigenstate of $\tau^z_{i-\frac{1}{2}}\sigma^x_i\tau^z_{i+\frac{1}{2}}$ for all $i$,

$$\tau^z_{i-\frac{1}{2}}\sigma^x_i\tau^z_{i+\frac{1}{2}}|\psi\rangle_{\text{PBC}} = |\psi\rangle_{\text{PBC}}, i = 1, ..., L-1, \qquad \tau^z_{L-\frac{1}{2}}\sigma^x_L\tau^z_{\frac{1}{2}}|\psi\rangle_{\text{PBC}} = |\psi\rangle_{\text{PBC}}. \tag{4.9}$$

In particular, this means that $|\psi\rangle_{\mathrm{PBC}}$ is neutral under $\mathbb{Z}_2^\sigma$,

$$U_\sigma |\psi\rangle_{\mathrm{PBC}} = \prod_{i=1}^{L} \sigma_i^x |\psi\rangle_{\mathrm{PBC}} = \left(\prod_{i=1}^{L-1} \tau_{i-\frac{1}{2}}^z \tau_{i+\frac{1}{2}}^z\right) \tau_{L-\frac{1}{2}}^z \tau_{\frac{1}{2}}^z |\psi\rangle_{\mathrm{PBC}} = |\psi\rangle_{\mathrm{PBC}}. \qquad (4.10)$$

However, the above method does not fix the $\mathbb{Z}_2^\tau$ charge of the ground state. By exact diagonalization, we confirmed that the ground state is $\mathbb{Z}_2^\tau$ even. Furthermore, exact diagonalization also shows that there is only one ground state under PBC, which is the desired property of gapless SPT [4].

We proceed to the TBC of $\mathbb{Z}_2^\tau$. The ground state $|\psi\rangle_{\mathrm{TBC}_\tau}$ satisfies

$$\tau_{i-\frac{1}{2}}^z \sigma_i^x \tau_{i+\frac{1}{2}}^z |\psi\rangle_{\mathrm{TBC}_\tau} = |\psi\rangle_{\mathrm{TBC}_\tau}, i = 1, ..., L-1, \qquad \tau_{L-\frac{1}{2}}^z \sigma_L^x \tau_{\frac{1}{2}}^z |\psi\rangle_{\mathrm{TBC}_\tau} = -|\psi\rangle_{\mathrm{TBC}_\tau}. \quad (4.11)$$

This means that $|\psi\rangle_{\mathrm{TBC}_\tau}$ is $\mathbb{Z}_2^\sigma$ odd,

$$U_\sigma |\psi\rangle_{\mathrm{TBC}_\tau} = \prod_{i=1}^{L} \sigma_i^x |\psi\rangle_{\mathrm{TBC}_\tau} = -\left(\prod_{i=1}^{L-1} \tau_{i-\frac{1}{2}}^z \tau_{i+\frac{1}{2}}^z\right) \tau_{L-\frac{1}{2}}^z \tau_{\frac{1}{2}}^z |\psi\rangle_{\mathrm{TBC}_\tau} = -|\psi\rangle_{\mathrm{TBC}_\tau}. \qquad (4.12)$$

One can again numerically check that $|\psi\rangle_{\mathrm{TBC}_\tau}$ is even under $\mathbb{Z}_2^\tau$.

We finally consider the TBC of $\mathbb{Z}_2^\sigma$. The two Hamiltonians under PBC and TBC of $\mathbb{Z}_2^\sigma$ are related by conjugating by $\tau_{\frac{1}{2}}^z$,

$$H_{\mathrm{gSPT}}^{\mathrm{TBC}_\sigma} = \tau_{\frac{1}{2}}^z H_{\mathrm{gSPT}}^{\mathrm{PBC}} \tau_{\frac{1}{2}}^z. \qquad (4.13)$$

Hence their ground states are also related,

$$|\psi\rangle_{\mathrm{TBC}_\sigma} = \tau_{\frac{1}{2}}^z |\psi\rangle_{\mathrm{PBC}}. \qquad (4.14)$$

As a consequence, the $\mathbb{Z}_2^\sigma$ charge of $|\psi\rangle_{\mathrm{TBC}_\sigma}$ and $|\psi\rangle_{\mathrm{PBC}}$ are the same, while their $\mathbb{Z}_2^\tau$ charge are the opposite.

As we see from the above, the discussion depends heavily on the form of the Hamiltonian. In particular, we repeatedly used the fact that the first term in the Hamiltonian (4.3) commutes with the rest of the terms. This will be no longer true if one adds a generic symmetric perturbation, for example

$$-h \sum_{i=1}^{L} \tau_{i-\frac{1}{2}}^x. \qquad (4.15)$$

In this situation, the analysis in the current subsection does not work, and one has to apply numerical computation to find the ground state charge. However, in Section 4.4, we will re-derive the above results using the KT transformation, and the result holds under perturbation as well hence is more powerful.

### 4.3.2 Degeneracy under open boundary condition

We proceed to discuss the topological features under OBC. There are many different open boundary conditions, depending on how one truncates the lattice, and what types of local interactions are added to the boundary. For simplicity, we focus on one particular boundary condition, where only the sites $i$ and $i - \frac{1}{2}$ for $i = 1, ..., L$ belong to the lattice. The Hamiltonian is chosen such that only the terms fully supported on the lattice are preserved. The Hamiltonian is

$$H_{\text{gSPT}}^{\text{OBC}} = -\sum_{i=1}^{L-1} \tau_{i-\frac{1}{2}}^z \sigma_i^x \tau_{i+\frac{1}{2}}^z - \sum_{i=2}^{L} \sigma_{i-1}^z \tau_{i-\frac{1}{2}}^x \sigma_i^z - \sum_{i=1}^{L} \sigma_i^x. \tag{4.16}$$

Then it is easy to check that the following terms commute with the Hamiltonian

$$\tau_{\frac{1}{2}}^z, \qquad \tau_{L-\frac{1}{2}}^z \sigma_L^x, \qquad U_\sigma, \qquad U_\tau. \tag{4.17}$$

The first two terms are localized on the boundaries, and the last two terms are symmetry operators. Because $\{\tau_{\frac{1}{2}}^z, U_\tau\} = \{\tau_{L-\frac{1}{2}}^z \sigma_L^x, U_\tau\} = 0$, the irreducible representation of the algebra is two dimensional. Hence there are two degenerate ground states.

Under the bulk perturbation (4.15), the boundary terms $\tau_{\frac{1}{2}}^z, \tau_{L-\frac{1}{2}}^z \sigma_L^x$ no longer commute with the Hamiltonian, and the degeneracy from the above are lifted. However, by perturbation theory analysis, the gap between the two lowest states decays exponentially with respect to the system size (See Section 2.4.1 in [4] for further details). This exponential edge degeneracy of gSPTs is also discussed by the decorated domain wall argument in references [2, 6].

## 4.4 Topological features from KT transformation

In this subsection, we reproduce the results in Section 4.3 using the KT transformation. We first analyze the topological features of the decoupled system (4.1) as well as its perturbations. Since the decoupled system is relatively simple, we know the symmetry properties even under perturbation. We then use the KT transformation to relate the symmetry properties of the decoupled system (4.1) to the gapless SPT (4.3). This will enable us to determine the symmetry properties of the ground states even after perturbation.

We first study the symmetry properties of the ground states before the KT transformation, i.e. (4.1), with a symmetric perturbation $-h \sum_{i=1}^{L} \tau_{i-\frac{1}{2}}^x$. We will assume $h \ll 1$ in this subsection. Hence the Hamiltonian is simply a decoupled critical Ising Hamiltonian plus a transverse field Ising model with a small transverse field (hence in deep SSB phase),

$$H_{\text{Ising+SSB+pert}} = H_{\text{Ising}} + H_{\text{SSB+pert}} \tag{4.18}$$

where

$$H_{\text{Ising}} = -\sum_{i=1}^{L} \left( \sigma_{i-1}^z \sigma_i^z + \sigma_i^x \right), \qquad H_{\text{SSB+pert}} = -\sum_{i=1}^{L} \left( \tau_{i-\frac{1}{2}}^z \tau_{i+\frac{1}{2}}^z + h \tau_{i-\frac{1}{2}}^x \right). \tag{4.19}$$

The symmetry properties of the ground states of the above two models are well-known. Let us denote the ground state energy of the critical Ising model $H_{\text{Ising}}$ as $E^{\sigma}_{(u_\sigma, t_\sigma)}$. It is well-known that they satisfy the following relations

$$E^{\sigma}_{(0,0)} \overset{\frac{1}{L}}{<} E^{\sigma}_{(1,0)} = E^{\sigma}_{(0,1)} \overset{\frac{1}{L}}{<} E^{\sigma}_{(1,1)}. \tag{4.20}$$

The symbol $E_1 \overset{\frac{1}{L}}{<} E_2$ means that the difference between the energies on its two sides $E_2 - E_1$ is of order $\frac{1}{L}$. The equality $E^{\sigma}_{(1,0)} = E^{\sigma}_{(0,1)}$ is ensured by the Kramer-Wannier self-duality of the Ising model where the KW exchanges $(u_\sigma, t_\sigma) \leftrightarrow (t_\sigma, u_\sigma)$.

The low energy spectrum of the SSB phase is also well-known. When $h = 0$, there are two exactly degenerate ground states under PBC, $|u_\tau = 0, 1\rangle$. When $h > 0$, the degeneracy is lifted, and where the gap between $\frac{1}{\sqrt{2}}(|u_\tau = 0\rangle + |u_\tau = 1\rangle)$ and $\frac{1}{\sqrt{2}}(|u_\tau = 0\rangle - |u_\tau = 1\rangle)$ decays exponentially with respect to the system size. Moreover, the ground state energy in the twisted sector is roughly the energy of the domain wall excitation, which is of order 1. Denote the ground state energy of the model $H_{\text{SSB+pert}}$ as $E^{\tau}_{(u_\tau, t_\tau)}$, then they satisfy the relation

$$E^{\tau}_{(0,0)} \overset{e^{-L}}{<} E^{\tau}_{(1,0)} \overset{1}{<} E^{\tau}_{(0,1)} \overset{\frac{1}{L^2}}{<} E^{\tau}_{(1,1)}. \tag{4.21}$$

See Appendix A for a concrete derivation using Jordan Wigner transformation.

We proceed to perform the KT transformation on (4.18). Since $\tau^{x}_{i-\frac{1}{2}}$ is mapped to itself, i.e. $\mathcal{N}_{\text{KT}} \tau^{x}_{i-\frac{1}{2}} = \tau^{x}_{i-\frac{1}{2}} \mathcal{N}_{\text{KT}}$, the perturbation in $-h\sum_{i=1}^{L} \tau^{x}_{i-\frac{1}{2}}$ is preserved under KT transformation. Hence we get the gapless SPT with perturbation (4.15),

$$H_{\text{gSPT+pert}} = -\sum_{j=1}^{L} \left( \tau^{z}_{i-\frac{1}{2}} \sigma^{x}_{i} \tau^{z}_{i+\frac{1}{2}} + \sigma^{z}_{i-1} \tau^{x}_{i-\frac{1}{2}} \sigma^{z}_{i} + \sigma^{x}_{i} + h\tau^{x}_{i-\frac{1}{2}} \right). \tag{4.22}$$

Denote the ground state energy of the Hamiltonian (4.22) in the symmetry-twist sector as $E^{\text{gSPT}}_{[(u_\sigma, t_\sigma),(u_\tau, t_\tau)]}$. By (2.6) and using $E^{\sigma}_{(u_\sigma, t_\sigma)}$ and $E^{\tau}_{(u_\tau, t_\tau)}$, we obtain

$$E^{\text{gSPT}}_{[(u_\sigma, t_\sigma),(u_\tau, t_\tau)]} = E^{\sigma}_{(u_\sigma, t_\sigma + u_\tau)} + E^{\tau}_{(u_\tau, t_\tau + u_\sigma)} \tag{4.23}$$

from which we are able to determine the charge of the ground state in each twist sector. Let us discuss them case by case.

1. $t_\sigma = 0, t_\tau = 0$: Both sigma and tau spins obey PBC. The energy (4.23) reduces to

$$
\begin{aligned}
E^{\text{gSPT}}_{[(u_\sigma,0),(u_\tau,0)]} &= E^{\sigma}_{(u_\sigma, u_\tau)} + E^{\tau}_{(u_\tau, u_\sigma)} \\
&= E^{\sigma}_{(0,0)} + E^{\tau}_{(0,0)} +
\begin{cases}
0, & (u_\sigma, u_\tau) = (0,0) \\
\frac{1}{L} + 1, & (u_\sigma, u_\tau) = (1,0) \\
\frac{1}{L} + e^{-L}, & (u_\sigma, u_\tau) = (0,1) \\
\frac{1}{L} + \frac{1}{L} + 1 + \frac{1}{L^2}. & (u_\sigma, u_\tau) = (1,1)
\end{cases}
\end{aligned} \tag{4.24}
$$

The minimal energy is achieved in the symmetry sector $(u_\sigma, u_\tau) = (0,0)$.

2. $t_\sigma = 1, t_\tau = 0$: The sigma spins obey TBC, and tau spins obey PBC. The energy (4.23) reduces to

$$E^{\text{gSPT}}_{[(u_\sigma,1),(u_\tau,0)]} = E^\sigma_{(u_\sigma,1+u_\tau)} + E^\tau_{(u_\tau,u_\sigma)}$$

$$= E^\sigma_{(0,0)} + E^\tau_{(0,0)} + \begin{cases} \frac{1}{L}, & (u_\sigma, u_\tau) = (0,0) \\ \frac{1}{L} + \frac{1}{L} + 1, & (u_\sigma, u_\tau) = (1,0) \\ e^{-L}, & (u_\sigma, u_\tau) = (0,1) \\ \frac{1}{L} + 1 + \frac{1}{L^2}. & (u_\sigma, u_\tau) = (1,1) \end{cases} \tag{4.25}$$

The minimal energy is achieved in the symmetry sector $(u_\sigma, u_\tau) = (0,1)$.

3. $t_\sigma = 0, t_\tau = 1$: The sigma spins obey PBC, and tau spins obey TBC. The energy (4.23) reduces to

$$E^{\text{gSPT}}_{[(u_\sigma,0),(u_\tau,1)]} = E^\sigma_{(u_\sigma,u_\tau)} + E^\tau_{(u_\tau,1+u_\sigma)}$$

$$= E^\sigma_{(0,0)} + E^\tau_{(0,0)} + \begin{cases} 1, & (u_\sigma, u_\tau) = (0,0) \\ \frac{1}{L}, & (u_\sigma, u_\tau) = (1,0) \\ \frac{1}{L} + 1 + \frac{1}{L^2}, & (u_\sigma, u_\tau) = (0,1) \\ \frac{1}{L} + \frac{1}{L} + e^{-L}. & (u_\sigma, u_\tau) = (1,1) \end{cases} \tag{4.26}$$

The minimal energy is achieved in the symmetry sector $(u_\sigma, u_\tau) = (1,0)$.

4. $t_\sigma = 1, t_\tau = 1$: The sigma spins obey TBC, and tau spins obey TBC. The energy (4.23) reduces to

$$E^{\text{gSPT}}_{[(u_\sigma,1),(u_\tau,1)]} = E^\sigma_{(u_\sigma,1+u_\tau)} + E^\tau_{(u_\tau,1+u_\sigma)}$$

$$= E^\sigma_{(0,0)} + E^\tau_{(0,0)} + \begin{cases} \frac{1}{L} + 1, & (u_\sigma, u_\tau) = (0,0) \\ \frac{1}{L} + \frac{1}{L}, & (u_\sigma, u_\tau) = (1,0) \\ 1 + \frac{1}{L^2}, & (u_\sigma, u_\tau) = (0,1) \\ \frac{1}{L} + e^{-L}. & (u_\sigma, u_\tau) = (1,1) \end{cases} \tag{4.27}$$

The minimal energy is achieved in the symmetry sector $(u_\sigma, u_\tau) = (1,1)$.

In the above, we only write down the schematic scaling behavior of energy with respect to the system size $L$. In summary, under the $\mathbb{Z}_2^\sigma$ or $\mathbb{Z}_2^\tau$ twisted boundary condition, the ground state carries nontrivial $\mathbb{Z}_2^\tau$ or $\mathbb{Z}_2^\sigma$ charge respectively. These results are not only consistent with, but also significantly generalize the discussions in Section 4.3 since we also allow a perturbation $-h \sum_i \tau^x_{i-\frac{1}{2}}$ here and the method in Section 4.3 no longer applies.

Furthermore, let us focus on the OBC where the KT transformation is unitary. When $h \ll 1$, the $\tau$ spins that remain in the $\mathbb{Z}_2$ SSB phase exhibit a two-fold (exponential) degeneracy of the ground states. After the KT transformation, this implies that the gSPT has two-fold exponential "edge" degeneracy, which still survives under the perturbation.

Let us make some comments.

1. The key property that we are able to determine the symmetry properties after the perturbation is that before the KT transformation the system (after perturbations) is decoupled and we know its structure well. A generic perturbation typically mixes the $\sigma$ and $\tau$ degrees of freedom after undoing KT transformation, and we will need numerics.

2. Since the KT transformation implements a twisted gauging, i.e. $STS$, the qualitative feature such as the location of the phase transition in terms of the perturbation $h$ can not change. Before the KT transformation, turning on a small perturbation $h$ does not trigger a phase transition, hence the system after KT transformation is also stable under turning on a small $h$. This means that the gapless SPT (4.3) is stable under the small perturbation (4.15). [4] We will study the phase diagram in the following subsection.

3. Discrete gauging also does not change the existence of gapped sectors. Since the system before the KT transformation contains a gapped sector, i.e. the SSB associated with the $\tau$ spins, after the KT transformation, there is still a gapped sector, even after a small perturbation. The existence of the gapped sector in the gapless SPT has been emphasized in [2–4,6], which comes from the gapped degrees of freedom decorating the domain wall, and they are crucial in protecting the non-trivial topological properties of the gapless SPT.

4. In general, the gapped sector can be seen from the exponential decaying energy splitting of edge modes under OBC. However, it is very difficult to prove the stability of such exponential decaying behavior under symmetric perturbations. With the help of KT transformation, we provide an analytical proof of this stability for gSPTs (1.5) and igSPTs (1.7) in appendix B.[5]

## 4.5   Phase diagram

In Section 4.4, we discussed the topological features of the gapless SPT using the KT transformation, and also showed that the gapless SPT is stable under a small perturbation $h$. In this subsection, we would like to understand the structure of the phases when one increases $h$, and determine the phase diagram. We will end up commenting on the string order parameter.

**Phase diagram:**   Since the qualitative structure of the phase diagram is not affected by (twisted) gauging of finite groups, the phase diagram and the location of phase transitions can be inferred from the system before the KT transformation. Before the KT transformation, the system is simply

---

[4]We would like to emphasize that in [4], we used the Hamiltonian (4.4) as the gapless SPT. Although (4.3) and (4.4) share the same low energy spectrum before turning on (4.15), adding such a perturbation would make the low energy spectrum different since the first term in (4.3) does not commute with the perturbation. Hence the discussion for the stability of gapless SPT under a small perturbation no longer applies to the discussion in Section 2.4 in [4]. Indeed, as one of the referees of [4] pointed out, the perturbation there would gap out the gapless SPT.

[5]Although such degeneracy under OBC is stable under symmetric perturbation, the system can be driven to the SSB phase or gapped SPT phase when the bulk perturbation can open a gap for the gapless systems before KT transformation.

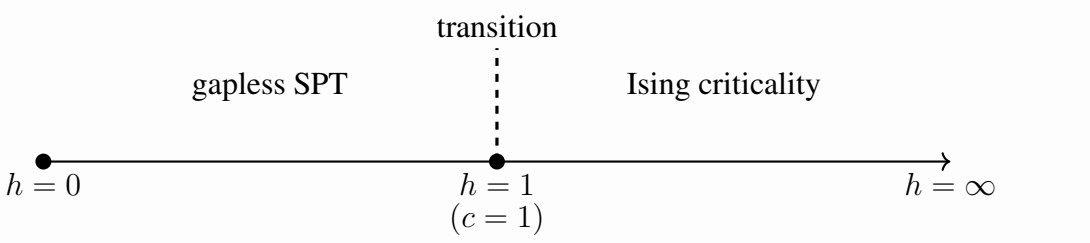

Figure 1: Phase diagram of the gapless SPT under perturbation.

a critical Ising model for $\sigma$ spin and a transverse field Ising model for $\tau$ spin, it is clear that there is only one phase transition at $h = 1$ and the total system is in the Ashkin-Teller (AT) university class [46,47]. Hence the gapless SPT is stable as long as the perturbation $h$ is smaller than 1.

Let us determine the symmetry properties of the ground state within each phase and at the phase transition. When $h < 1$, the symmetry properties of the ground state in different sectors have been discussed in Section 4.4. When $h > 1$, by using the same method as in Section 4.4, we find that after the KT transformation, the ground states under four boundary conditions are all $\mathbb{Z}_2^\sigma$ even and $\mathbb{Z}_2^\tau$ even. Indeed, in the Hamiltonian (4.22) in the large $h$ limit, the last term dominates, and the ground state satisfies $\tau_{i-\frac{1}{2}}^x \simeq 1$. Then the Hamiltonian simplifies to an Ising paramagnetic (trivially gapped phase) for the $\tau$ spin and a critical Ising model for the $\sigma$ spin. Indeed, in such a model, the ground state under any boundary condition is even for both $\mathbb{Z}_2^\sigma$ and $\mathbb{Z}_2^\tau$. We plot the phase diagram as in Figure 1. The system in the entire phase diagram is gapless. The transition is when the gapped sector becomes gapless in the gapless system.

The transition $h = 1$ is described by free boson CFT in low energy with central charge $c = 1$, while the central charge away from $h = 1$ is $c = \frac{1}{2}$. This phase transition is the KT dual theory of two decoupled lsing criticality. However, it is also a phase transition between the SPT and trivial phases, which is an anomalous theory [42] and is distinct from gSPT models constructed using decoupled free boson CFTs in Eq.(1.8) and Eq.(1.9) [6].

**String order parameter:** We finally comment on the string order parameter in the gapless SPT. We start with the local order parameter for the decoupled Hamiltonian (4.1) before the KT transformation,

$$\langle \sigma_i^z \sigma_j^z \rangle \sim \frac{1}{|i-j|^{2\Delta}}, \qquad \langle \tau_{i-\frac{1}{2}}^z \tau_{j-\frac{1}{2}}^z \rangle \sim \begin{cases} \mathcal{O}(1), & h < 1 \\ e^{-\xi L}, & h > 1 \end{cases} \tag{4.28}$$

where $\Delta$ is the scaling dimension of $\sigma^z$.

---

[6]Since the gapless SPT for $h < 1$ can be understood as stacking an Ising criticality with a gapped SPT phase, the transition at $h = 1$ can also be intuitively understood as an Ising criticality stacked with phase transition between the SPT and trivial phases. As lsing criticality is anomaly-free, the anomaly of the transition at $h = 1$ only comes from phase transition between the SPT and trivial phases

After the KT transformation, the local order parameters become string order parameters and obey the same scaling behavior,

$$\langle \sigma_i^z (\prod_{k=i}^{j-1} \tau_{k+\frac{1}{2}}^x)\sigma_j^z \rangle \sim \frac{1}{|i-j|^{2\Delta}}, \qquad \langle \tau_{i-\frac{1}{2}}^z (\prod_{k=i}^{j-1} \sigma_k^x)\tau_{j-\frac{1}{2}}^z \rangle \sim \begin{cases} \mathcal{O}(1), & h < 1 \\ e^{-\xi L}. & h > 1 \end{cases} \qquad (4.29)$$

# 5 Intrinsically gapless SPT from KT transformation

In Section 4, we constructed the gapless SPT by applying KT transformation on a decoupled critical Ising model stacked with a $\mathbb{Z}_2$ SSB phase. In this section, we will apply the KT transformation to construct the intrinsically gapless SPT (igSPT) which was first found in [3], and later revisited in [4, 5].

The intrinsically gapless SPT states is a class of gapless systems which exhibit an emergent anomaly of the low energy symmetries. Although the entire global symmetry $G$ is anomaly free, $G$ does not act faithfully on the low energy gapless degrees of freedom. There is a normal subgroup $H$ of $G$ which only acts on the gapped sector, hence the quotient $G/H$ acts faithfully on the low energy sector. Because $G$ is a nontrivial extension of $G/H$ by $H$, $G/H$ then has a nontrivial 't Hooft anomaly [11, 48, 49], named the emergent anomaly in [3]. In [4], building upon [3, 11], the gapped sector was rephrased in terms of the anomalous SPT in the modified decorated domain wall construction. The emergent anomaly of $G/H$ is canceled by the anomalous SPT with $H$ symmetry, hence the total symmetry $G$ is anomaly free. The gapped sector turns out to be crucial to protect the nontrivial topological properties of the intrinsically gapless SPT.

In this section, we will construct an example of intrinsically gapless SPT with $G = \mathbb{Z}_4$ and $H = \mathbb{Z}_2$, which was originally discussed in [4]. We will find that this can be achieved by starting with a decoupled stacking of XX chain with a $\mathbb{Z}_2$ SSB phase, and then performing the KT transformation. The KT transformation also allows us to analytically determine the phase transition under the perturbation of igSPT, which we were unable to determine by small-scale numerical calculation in [4].

## 5.1 Constructing the intrinsically gapless SPT

Let us start with an Ising SSB Hamiltonian stacked with an XX chain. The Hamiltonian is

$$H_{\text{SSB+XX}} = -\sum_{i=1}^{L} \left( \tau_{i-\frac{1}{2}}^z \tau_{i+\frac{1}{2}}^z + \tau_{i-\frac{1}{2}}^y \tau_{i+\frac{1}{2}}^y + \sigma_{i-1}^z \sigma_i^z \right). \qquad (5.1)$$

This Hamiltonian actually has a larger $U(1)^\tau \times \mathbb{Z}_2^\sigma$ global symmetry, where $\mathbb{Z}_2^\sigma$ is generated by $U_\sigma$ defined in (2.2), and $U(1)^\tau$ is generated by

$$\prod_{i=1}^{L} e^{\frac{i\alpha}{2}(1-\tau_{i-\frac{1}{2}}^x)}, \qquad \alpha \simeq \alpha + 2\pi. \qquad (5.2)$$

The $\mathbb{Z}_2^\tau$ normal subgroup of $U(1)^\tau$ is generated by $U_\tau$ in (2.2). We will instead consider the $\mathbb{Z}_4^\tau$ subgroup of $U(1)^\tau$, where the generator of $\mathbb{Z}_4^\tau$ is

$$V_\tau = \prod_{i=1}^{L} e^{\frac{i\pi}{4}(1-\tau_{i-\frac{1}{2}}^x)} \tag{5.3}$$

satisfying $V_\tau^2 = U_\tau$. We will be interested in the $\mathbb{Z}_4^\tau \times \mathbb{Z}_2^\sigma$ symmetry, generated by $V_\tau$ and $U_\sigma$.

Let us apply the KT transformation, by using the $\mathbb{Z}_2^\tau \times \mathbb{Z}_2^\sigma$ symmetry generated by $U_\tau$ and $U_\sigma$. Under KT transformation, we have

$$\begin{aligned}
\mathcal{N}_{\text{KT}} \tau_{i-\frac{1}{2}}^z \tau_{i+\frac{1}{2}}^z &= \tau_{i-\frac{1}{2}}^z \sigma_i^x \tau_{i+\frac{1}{2}}^z \mathcal{N}_{\text{KT}}, \\
\mathcal{N}_{\text{KT}} \tau_{i-\frac{1}{2}}^y \tau_{i+\frac{1}{2}}^y &= \tau_{i-\frac{1}{2}}^y \sigma_i^x \tau_{i+\frac{1}{2}}^y \mathcal{N}_{\text{KT}}, \\
\mathcal{N}_{\text{KT}} \sigma_{i-1}^z \sigma_i^z &= \sigma_{i-1}^z \tau_{i-\frac{1}{2}}^x \sigma_i^z \mathcal{N}_{\text{KT}}.
\end{aligned} \tag{5.4}$$

Hence the Hamiltonian after KT transformation is precisely the intrinsically gapless SPT (or strong SPTC) found in [4]

$$H_{\text{igSPT}} = -\sum_{i=1}^{L} \left( \tau_{i-\frac{1}{2}}^z \sigma_i^x \tau_{i+\frac{1}{2}}^z + \tau_{i-\frac{1}{2}}^y \sigma_i^x \tau_{i+\frac{1}{2}}^y + \sigma_{i-1}^z \tau_{i-\frac{1}{2}}^x \sigma_i^z \right). \tag{5.5}$$

Since the KT transformation commutes with both $\sigma_i^x$ and $\tau_{i-\frac{1}{2}}^x$, the resulting Hamiltonian (5.5) also has $\mathbb{Z}_4^\tau \times \mathbb{Z}_2^\sigma$ symmetry, generated by $V_\tau$ and $U_\sigma$ respectively.

In [4], the Hamiltonian (5.5) was claimed to be the intrinsically gapless SPT protected by $\mathbb{Z}_4^\Gamma$. This $\mathbb{Z}_4^\Gamma$ is generated by the product $U_\sigma V_\tau$. Here, we would like to emphasize that not only the combination $U_\sigma V_\tau$ commutes with the Hamiltonian, but both $U_\sigma$ and $V_\tau$ separately commutes with (5.5) as well. This accidental symmetry allows us to construct it using the KT transformation.

## 5.2 Mapping between symmetry-twist sectors

Since the intrinsic gapless SPT has an accidental anomaly-free symmetry $\mathbb{Z}_4^\tau \times \mathbb{Z}_2^\sigma$, we first consider the symmetry and twist sectors. The symmetry sectors are labeled by the eigenvalues of the operator $V_\tau$ and $U_\sigma$, which are $e^{\frac{i\pi}{2} v_\tau}$ for $v_\tau = 0, 1, 2, 3$ and $(-1)^{u_\sigma}$ for $u_\sigma = 0, 1$ respectively. The twist sector (i.e. the boundary condition) is defined by

$$\begin{aligned}
|s_{i+L}^\sigma\rangle &= \sum_{s_i^\sigma = 0,1} \left[ (\sigma_i^x)^{t_\sigma} \right]_{s_{i+L}^\sigma, s_i^\sigma} |s_i^\sigma\rangle = |s_i^\sigma + t_\sigma\rangle \\
|s_{i-\frac{1}{2}+L}^\tau\rangle &= \sum_{s_{i-\frac{1}{2}}^\tau = 0,1} \left[ e^{\frac{i\pi r_\tau}{4}(1-\tau_{i-\frac{1}{2}}^x)} \right]_{s_{i-\frac{1}{2}+L}^\tau, s_{i-\frac{1}{2}}^\tau} |s_{i-\frac{1}{2}}^\tau\rangle = \frac{1}{2} \sum_{s_{i-\frac{1}{2}}^\tau = 0,1} \left( 1 + e^{\frac{i\pi}{2} r_\tau + i\pi(s_{i-\frac{1}{2}}^\tau + s_{i-\frac{1}{2}+L}^\tau)} \right) |s_{i-\frac{1}{2}}^\tau\rangle
\end{aligned} \tag{5.6}$$

where $t_\sigma \simeq t_\sigma + 2$, and $r_\tau \simeq r_\tau + 4$. In particular, when $r_\tau = 2t_\tau$, the second equality in the above simplifies to $|s_{i-\frac{1}{2}+L}^\tau\rangle = |s_{i-\frac{1}{2}}^\tau + t_\tau\rangle$. Hence the symmetry and twist sectors are labeled by

$$[(u_\sigma, t_\sigma), (v_\tau, r_\tau)], \qquad u_\sigma, t_\sigma \in \mathbb{Z}_2, \qquad v_\tau, r_\tau \in \mathbb{Z}_4. \tag{5.7}$$

To show how the $\mathbb{Z}_4^\tau \times \mathbb{Z}_2^\sigma$ symmetry and twist sectors are mapped under the KT transformation, we need to duplicate the same discussion as in Section 6.2 of [1]. However, as we know that the KT transformation implements the twisted gauging $STS$, it turns out that it is much easier to obtain the map from the partition function directly. Relegating the derivation to the Appendix C, we find the mapping between symmetry and twist sectors as

$$[(u'_\sigma, t'_\sigma), (v'_\tau, r'_\tau)] = [(u_\sigma, t_\sigma + v_\tau), (v_\tau, r_\tau + 2u_\sigma)]. \tag{5.8}$$

Indeed, the symmetry sectors $u_\sigma$ and $v_\tau$ are unchanged under the KT transformation, which directly follows from the observation below (5.5) that the symmetry operators $U_\sigma$ and $V_\tau$ are unchanged under the KT transformation.

Note that the intrinsic gapless SPT is protected by $\mathbb{Z}_4^\Gamma$, rather than $\mathbb{Z}_2^\sigma \times \mathbb{Z}_4^\tau$. Since the gapless SPTs appear after the KT transformation, we denote the symmetry-twist sectors of $\mathbb{Z}_4^\Gamma$ by $(u'_\Gamma, t'_\Gamma)$ where the prime stands for the sectors after the KT transformation according to (5.8). Note that $U_\Gamma = U_\sigma V_\tau$, their eigenvalues are thus related by $e^{\frac{i\pi}{2}u'_\Gamma} = (-1)^{u'_\sigma} e^{\frac{i\pi}{2}v'_\tau}$. Hence

$$u'_\Gamma = 2u'_\sigma + v'_\tau \mod 4. \tag{5.9}$$

To see how the twist sectors are related, we see that $\mathbb{Z}_4^\Gamma$ twisted boundary condition is determined by

$$|\sigma_{i+L}^\sigma, s_{i-\frac{1}{2}+L}^\tau\rangle = \sum_{s_i^\sigma, s_{i-\frac{1}{2}}^\tau = 0,1} \left[(\sigma_i^x)^{t'_\Gamma}\right]_{s_{i+L}^\sigma, s_i^\sigma} \left[e^{\frac{i\pi t'_\Gamma}{4}(1-\tau_{i-\frac{1}{2}}^x)}\right]_{s_{i-\frac{1}{2}+L}^\tau, s_{i-\frac{1}{2}}^\tau} |\sigma_i^\sigma, s_{i-\frac{1}{2}}^\tau\rangle. \tag{5.10}$$

Comparing with (5.6), we find

$$t'_\sigma = t'_\Gamma \mod 2, \qquad r'_\tau = t'_\Gamma \mod 4. \tag{5.11}$$

Indeed, the $\mathbb{Z}_2^\sigma \times \mathbb{Z}_4^\tau$ charge completely determines the $\mathbb{Z}_4^\Gamma$ charge. However, not every $\mathbb{Z}_2^\sigma \times \mathbb{Z}_4^\tau$ twist sector gives rise to a consistent $\mathbb{Z}_4^\Gamma$ twist sector. (5.11) implies that a consistent $\mathbb{Z}_4^\Gamma$ twist sector exists only when $t'_\sigma = r'_\tau \mod 2$.

How are the $\mathbb{Z}_4^\Gamma$ symmetry-twist sectors determined in terms of the sectors before the KT transformation? From (5.9) and (5.11), we find that the $\mathbb{Z}_2^\sigma \times \mathbb{Z}_4^\tau$ symmetry and twist sectors before KT the transformation are related to the $\mathbb{Z}_4^\Gamma$ symmetry and twist sectors after the KT transformation as

$$(u'_\Gamma, t'_\Gamma) = (2u'_\sigma + v'_\tau, r'_\tau) = (2u_\sigma + v_\tau, r_\tau + 2u_\sigma). \tag{5.12}$$

From the previous paragraph, the first equality in (5.12) holds only when $t'_\sigma = r'_\tau \mod 2$, or equivalently $t_\sigma = v_\tau + r_\tau \mod 2$. Note that the twist parameter $r_\tau + 2u_\sigma$ on the right hand side of (5.12) can not be consistently written in terms of a $\mathbb{Z}_4^\Gamma$ twist. This means that in order to obtain consistent $\mathbb{Z}_4^\Gamma$ untwisted and twisted sectors after KT transformation, we should consider the entire $\mathbb{Z}_2^\sigma \times \mathbb{Z}_4^\Gamma$ untwisted and twisted sectors with the $\mathbb{Z}_2^\sigma$ twist parameter fixed by $t_\sigma = v_\tau + r_\tau$ mod 2 before the KT transformation, rather than the $\mathbb{Z}_4^\Gamma$ untwisted and twisted sectors before the KT transformation. In short, the $\mathbb{Z}_4^\Gamma$ (un)twisted sectors are not invariant under KT transformation.

## 5.3 Field theory of the intrinsically gapless SPT

The KT transformation also allows us to write down the field theory for the intrinsically gapless SPT. We start with the partition function for the free boson CFT + SSB phase as the low energy theory of Eq. (5.1) [50]

$$Z_{\text{free boson}}[A_\tau]Z_{\text{SSB}}[A_\sigma], \tag{5.13}$$

where the partition function of the free boson CFT is given by

$$Z_{\text{free boson}}[A_\tau] := \int \mathcal{D}\theta \exp\left(i\int_{X_2} \frac{1}{2\pi}(D_{A_\tau}\theta)^2\right), \qquad D_{A_\tau}\theta = d\theta - i\frac{\pi}{2}A_\tau\theta. \tag{5.14}$$

Here $A_\tau$ is a $\mathbb{Z}_4$ 1-cocycle. And the partition function of the $\mathbb{Z}_2^\tau$ SSB phase is also simply a delta function,

$$Z_{\text{SSB}}[A_\sigma] = \delta(A_\sigma). \tag{5.15}$$

We then perform a $STS$ transformation, changing the partition function to

$$\begin{aligned}
Z_{\text{igSPT}}[A_\sigma, A_\tau] &= \sum_{\substack{a_\sigma=0,1, \\ a_\tau=A_\tau \mod 2}} Z_{\text{SSB}}[a_\sigma]Z_{\text{free boson}}[a_\tau]e^{\frac{i\pi}{2}\int_{X_2}(A_\tau-a_\tau)(A_\sigma+a_\sigma)} \\
&= \sum_{a_\tau=A_\tau \mod 2} Z_{\text{free boson}}[a_\tau]e^{\frac{i\pi}{2}\int_{X_2}(A_\tau A_\sigma-a_\tau A_\sigma)}
\end{aligned} \tag{5.16}$$

where the detail of derivation is shown in appendix C. Finally, the igSPT in [4] was defined with respect to the symmetry $\mathbb{Z}_4^\Gamma$. Denoting the $\mathbb{Z}_4^\Gamma$ background field as $A_\Gamma$, from (5.11), we identify $A_\tau = A_\Gamma \mod 4, A_\sigma = A_\Gamma \mod 2$. The partition function is then

$$Z_{\text{igSPT}}[A_\Gamma] = \sum_{a=A_\Gamma \mod 2} Z_{\text{free boson}}[a]e^{\frac{i\pi}{2}\int_{X_2}(A_\Gamma-a)A_\Gamma}. \tag{5.17}$$

We proceed to see the relation between (5.17) and the decorated domain wall construction in [3, 4]. To see this, we decompose the $\mathbb{Z}_4^\Gamma$ background field $A_\Gamma$ into $A_\Gamma = 2B + A$ with the constraint $\delta B = A^2 \mod 2$, and the right hand side of (5.17) can be precisely factorized into the form of Eq.(6) in [3],

$$Z_{\text{igSPT}}[A_\Gamma] = Z_{\text{low}}[A]Z_{\text{gapped}}[A, B] \tag{5.18}$$

where

$$Z_{\text{low}}[A] = \sum_{b=0,1,\delta b=A^2} Z_{\text{free boson}}[2b + A]e^{i\pi\int_{X_2} bA}, \qquad Z_{\text{gapped}}[A, B] = e^{i\pi\int_{X_2} AB}. \tag{5.19}$$

Note that the low energy partition function $Z_{\text{low}}[A]$ depends only on the quotient $\mathbb{Z}_2$ background field, and the term $e^{i\pi\int_{X_2} BA}$ plays the role of anomalous domain wall decoration. Note that both $Z_{\text{low}}[A]$ and $Z_{\text{gapped}}[A, B]$ are anomalous. The former is anomalous due to the constraint $\delta b = A^2$ and the latter is due to the constraint $\delta B = A^2$.

## 5.4 Topological features of intrinsically gapless SPT

We proceed to study the topological features of the intrinsically gapless SPT directly from the Hamiltonian (5.5). The content of this subsection already appeared in Section 3 of [4], which we briefly review here. We will focus on $\mathbb{Z}_4^\Gamma$ symmetry charge of the ground state under TBC of $\mathbb{Z}_2^\tau$ and $\mathbb{Z}_4^\Gamma$. Note that the $\mathbb{Z}_4^\Gamma$ symmetry charge operator is given by the product $U_\sigma V_\tau$.

As we found in [4], under PBC, the number of ground states depends on the number of sites. This is potentially due to the effective twisted boundary condition when the system size is of a certain type. We will focus on the sequence of system size where the ground state is unique under PBC. We also found that the $\mathbb{Z}_4^\Gamma$ charge of the ground state depends on the system size as well, even when the ground state is unique. However, the $\mathbb{Z}_2^\tau$ normal subgroup of $\mathbb{Z}_4^\Gamma$, generated by $U_\tau$, is always trivial. This can be seen as follows. Note that the last term in (5.5) commutes with the first two terms, the ground state $|\psi\rangle_{\text{PBC}}$ must be an eigenstate of each of them,

$$\sigma_{i-1}^z \tau_{i-\frac{1}{2}}^x \sigma_i^z |\psi\rangle_{\text{PBC}} = |\psi\rangle_{\text{PBC}}, i = 2, ..., L, \qquad \sigma_L^z \tau_{\frac{1}{2}}^x \sigma_1^z |\psi\rangle_{\text{PBC}} = |\psi\rangle_{\text{PBC}}. \tag{5.20}$$

In particular, this means that $|\psi\rangle_{\text{PBC}}$ is neutral under $\mathbb{Z}_2^\tau$ normal subgroup of $\mathbb{Z}_4^\tau$,

$$U_\tau |\psi\rangle_{\text{PBC}} = \prod_{i=1}^{L} \tau_{i-\frac{1}{2}}^x |\psi\rangle_{\text{PBC}} = \left( \prod_{i=2}^{L} \sigma_{i-1}^z \sigma_i^z \right) \sigma_L^z \sigma_1^z |\psi\rangle_{\text{PBC}}. \tag{5.21}$$

We proceed to the TBC of $\mathbb{Z}_2^\tau$. We first consider twisting by $\mathbb{Z}_2^\tau$ normal subgroup. By restricting all the Pauli operators within the physical lattice, i.e. $i = 1, ..., L$, we find

$$
\begin{aligned}
H_{\text{igSPT}}^{\mathbb{Z}_2^\tau} &= -\sum_{i=1}^{L-1} \left( \tau_{i-\frac{1}{2}}^z \sigma_i^x \tau_{i+\frac{1}{2}}^z + \tau_{i-\frac{1}{2}}^y \sigma_i^x \tau_{i+\frac{1}{2}}^y + \sigma_i^z \tau_{i+\frac{1}{2}}^x \sigma_{i+1}^z \right) + \tau_{L-\frac{1}{2}}^z \sigma_L^x \tau_{\frac{1}{2}}^z + \tau_{L-\frac{1}{2}}^y \sigma_L^x \tau_{\frac{1}{2}}^y - \sigma_L^z \tau_{\frac{1}{2}}^x \sigma_1^z \\
&= \sigma_L^z H_{\text{igSPT}} \sigma_L^z.
\end{aligned}
\tag{5.22}
$$

The above relation immediately implies that the relative $\mathbb{Z}_4^\Gamma$ charge between the ground state under the $\mathbb{Z}_2^\tau$ TBC $|\psi\rangle_{\mathbb{Z}_2^\tau \text{TBC}}$ and the ground state under PBC $|\psi\rangle_{\text{PBC}}$ is 2 mod 4. Namely, we have

$$_{\mathbb{Z}_2^\tau \text{TBC}}\langle\psi| U_\Gamma |\psi\rangle_{\mathbb{Z}_2^\tau \text{TBC}} = -_{\text{PBC}} \langle\psi| U_\Gamma |\psi\rangle_{\text{PBC}}. \tag{5.23}$$

We finally consider the TBC of $\mathbb{Z}_4^\Gamma$, under which the Hamiltonian becomes

$$H_{\text{igSPT}}^{\mathbb{Z}_4^\Gamma} = -\sum_{i=1}^{L-1} \left( \tau_{i-\frac{1}{2}}^z \sigma_i^x \tau_{i+\frac{1}{2}}^z + \tau_{i-\frac{1}{2}}^y \sigma_i^x \tau_{i+\frac{1}{2}}^y + \sigma_i^z \tau_{i+\frac{1}{2}}^x \sigma_{i+1}^z \right) - \tau_{L-\frac{1}{2}}^z \sigma_L^x \tau_{\frac{1}{2}}^y + \tau_{L-\frac{1}{2}}^y \sigma_L^x \tau_{\frac{1}{2}}^z + \sigma_L^z \tau_{\frac{1}{2}}^x \sigma_1^z. \tag{5.24}$$

There are actually more than one ground state, but it is then straightforward to show that all the ground states $|\psi\rangle_{\mathbb{Z}_4^\Gamma \text{TBC}}$ carry $\mathbb{Z}_2^\tau$ charge 1 mod 2. By comparing the $\mathbb{Z}_2^\tau$ charge of the PBC ground state in (5.21), we again find that the relative charge between the $\mathbb{Z}_4^\Gamma$ TBC ground state and the PBC ground state is 1 mod 2, namely we have

$$_{\mathbb{Z}_4^\Gamma \text{TBC}}\langle\psi| U_\tau |\psi\rangle_{\mathbb{Z}_4^\Gamma \text{TBC}} = -_{\text{PBC}} \langle\psi| U_\tau |\psi\rangle_{\text{PBC}}. \tag{5.25}$$

Similar to the discussion in Section 4.3, the above discussion heavily depends on the form of the Hamiltonian. In particular, we repeatedly used the fact that the last term in the Hamiltonian (5.5) commutes with the remaining terms. Under a generic perturbation, for instance,

$$-h \sum_{i=1}^{L} \left( \sigma_i^x + \tau_{i-\frac{1}{2}}^x \right) \tag{5.26}$$

would destroy this feature. In this situation, the analysis in the current subsection does not work. In Section 3.4 of [4], we used small scale exact diagonalization to study the ground state charge under various boundary conditions as well as the energy gap as a function of perturbation (5.26). We found that when the perturbation $h$ is small, the relative symmetry charge discussed in the previous paragraphs are unchanged until $h$ increases to some critical value $h_c$. As $h$ further increases, the charges are then subjected to oscillations until around $h \simeq 2$, after which the relative charge becomes trivial. The value of $h_c$ changes with respect to the system size, and it was not clear the behavior of $h_c$ in the thermodynamics limit $L \to \infty$. Below, we will use the KT transformation to analytically study the phase diagram under the perturbation (5.26).

## 5.5 Topological features from the KT transformation

### 5.5.1 Symmetry properties before the KT transformation

In this subsection, we reproduce the results in Section 5.4 using the KT transformation. We first analyze the topological features of the decoupled system (5.1) as well as its perturbation (5.26). In this subsection, we will assume $h \ll 1$, and will discuss the phase diagram for finite $h$ in the next subsection. The Hamiltonian is

$$H_{\text{XX+SSB+pert}} = H_{\text{XX+pert}} + H_{\text{SSB+pert}} \tag{5.27}$$

where

$$
\begin{aligned}
H_{\text{XX+pert}} &= -\sum_{i=1}^{L} \left( \tau_{i-\frac{1}{2}}^z \tau_{i+\frac{1}{2}}^z + \tau_{i-\frac{1}{2}}^y \tau_{i+\frac{1}{2}}^y + h\tau_{i-\frac{1}{2}}^x \right), \\
H_{\text{SSB+pert}} &= -\sum_{i=1}^{L} \left( \sigma_{i-1}^z \sigma_i^z + h\sigma_i^x \right).
\end{aligned}
\tag{5.28}
$$

The symmetry properties of the ground states of the above two models are well-known. In Appendix D, we discuss the ground state properties of $H_{\text{XX+pert}}$, and find that the ground state energy in each symmetry-twist sector is

$$E_{(v_\tau, r_\tau)}^\tau = \frac{2\pi\sqrt{1 - h^2/4}}{L} \left[ \frac{1}{4} (\min([v_\tau]_4, 4 - [v_\tau]_4))^2 + \frac{1}{16} (\min([r_\tau]_4, 4 - [r_\tau]_4))^2 \right]. \tag{5.29}$$

Hence

$$
E^\tau_{(0,0)} \overset{\frac{1}{L}}{<} \begin{matrix} E^\tau_{(0,1)} \\ E^\tau_{(0,3)} \end{matrix} \overset{\frac{1}{L}}{<} \begin{matrix} E^\tau_{(1,0)} \\ E^\tau_{(3,0)} \\ E^\tau_{(0,2)} \end{matrix} \overset{\frac{1}{L}}{<} \begin{matrix} E^\tau_{(1,1)} \\ E^\tau_{(3,1)} \\ E^\tau_{(1,3)} \\ E^\tau_{(3,3)} \end{matrix} \overset{\frac{1}{L}}{<} \begin{matrix} E^\tau_{(1,2)} \\ E^\tau_{(3,2)} \end{matrix} \overset{\frac{1}{L}}{<} E^\tau_{(2,0)} \overset{\frac{1}{L}}{<} \begin{matrix} E^\tau_{(2,1)} \\ E^\tau_{(2,3)} \end{matrix} \overset{\frac{1}{L}}{<} E^\tau_{(3,3)} \tag{5.30}
$$

where the energies on each column are equal. The symmetry properties of $H_{\text{SSB+pert}}$ has been already considered in (4.21), which we reproduce here

$$
E^\sigma_{(0,0)} \overset{e^{-L}}{<} E^\sigma_{(1,0)} \overset{1}{<} E^\sigma_{(0,1)} \overset{\frac{1}{L^2}}{<} E^\sigma_{(1,1)}. \tag{5.31}
$$

The ground state energy of the Hamiltonian $H_{\text{XX+SSB+pert}}$ in the symmetry-twist sector $[(u_\sigma, t_\sigma), (v_\tau, r_\tau)]$ is given by

$$
E_{[(u_\sigma,t_\sigma),(v_\tau,r_\tau)]} = E^\tau_{(v_\tau,r_\tau)} + E^\sigma_{(u_\sigma,t_\sigma)}. \tag{5.32}
$$

### 5.5.2 Symmetry properties after the KT transformation

Under the KT transformation, the decoupled Hamiltonian (5.27) becomes the perturbation of igSPT,

$$
H_{\text{igSPT+pert}} = -\sum_{i=1}^{L} \left( \tau^z_{i-\frac{1}{2}} \sigma^x_i \tau^z_{i+\frac{1}{2}} + \tau^y_{i-\frac{1}{2}} \sigma^x_i \tau^y_{i+\frac{1}{2}} + \sigma^z_{i-1} \tau^x_{i-\frac{1}{2}} \sigma^z_i + h\sigma^x_i + h\tau^x_{i-\frac{1}{2}} \right). \tag{5.33}
$$

Note that the perturbation is the same as in the decoupled system (5.27) because $\sigma^x$ and $\tau^x$ commute with KT transformation. By combining the ground state symmetry properties of the decoupled system before gauging in Section 5.5.1 and how the symmetry-twist sectors are mapped under KT transformation in Section 5.2, we are able to determine the ground state properties of the perturbed igSPT Hamiltonian (5.33) after the KT transformation. In particular, we would like to determine the $\mathbb{Z}_4^\Gamma$ symmetry charge of the ground state under each $\mathbb{Z}_4^\Gamma$ twisted boundary condition.

Concretely, by (5.12), the energy in the symmetry-twist sector after the KT transformation $E^\Gamma_{(u_\Gamma, t_\Gamma)}$ is equal to the energy before the KT transformation $E_{[(u_\sigma,t_\sigma),(v_\tau,r_\tau)]}$ which is further equal to $E^\tau_{(v_\tau,r_\tau)} + E^\sigma_{(u_\sigma,t_\sigma)}$ by (5.32). Here, $(u_\Gamma, t_\Gamma)$ is determined by $[(u_\sigma, t_\sigma), (v_\tau, r_\tau)]$ via the relation $(u_\Gamma, t_\Gamma) = [(u_\sigma, t_\sigma), (v_\tau, r_\tau)]$. In summary, we have

$$
E^\Gamma_{(u_\Gamma,t_\Gamma)} \overset{(5.12)}{=} E_{[(u_\sigma,t_\sigma),(v_\tau,r_\tau)]} \overset{(5.32)}{=} E^\sigma_{(u_\sigma,t_\sigma)} + E^\tau_{(v_\tau,r_\tau)} \overset{(5.12)}{=} E^\sigma_{(u_\sigma,u_\Gamma+t_\Gamma)} + E^\tau_{(u_\Gamma+2u_\sigma,t_\Gamma+2u_\sigma)}. \tag{5.34}
$$

In the last equality, we used (5.12) to solve $v_\tau = u_\Gamma + 2u_\sigma \mod 4$ and $r_\tau = t_\Gamma + 2u_\sigma \mod 4$, and also used the condition $t_\sigma \overset{\mod 2}{=} v_\tau + r_\tau = u_\Gamma + t_\Gamma \mod 2$ which is imposed by (5.12). For a fixed twisted boundary condition $t_\Gamma$, we search for the minimal energy over all possible $u_\sigma$ and $u_\Gamma$ by comparing the energy hierarchy (5.30) and (5.31). The selected $u_\Gamma$ is the $\mathbb{Z}_4^\Gamma$ charge of the ground state in the $\mathbb{Z}_4^\Gamma$ twisted boundary condition specified by $t_\Gamma$. We will discuss the four twisted boundary conditions separately.

1. $t_\Gamma = 0 \mod 4$: By (5.34), the energy of the ground state in the symmetry-twist sector is

$$E^\Gamma_{(u_\Gamma, 0)} = E^\sigma_{(u_\sigma, u_\Gamma)} + E^\tau_{(u_\Gamma + 2u_\sigma, 2u_\sigma)}. \tag{5.35}$$

We would like to minimize the right hand side over all possible $u_\sigma$ and $u_\Gamma$. It is obvious that $u_\sigma = 0$ and $u_\Gamma = 0$ yield the lowest energy. We will only be interested in the relative symmetry charge between under TBC ($t_\Gamma = 1, 2, 3$) and PBC ($t_\Gamma = 0$).[7]

2. $t_\Gamma = 1 \mod 4$: By (5.34), the energy of the ground state in the symmetry-twist sector is

$$E^\Gamma_{(u_\Gamma, 1)} = E^\sigma_{(u_\sigma, u_\Gamma + 1)} + E^\tau_{(u_\Gamma + 2u_\sigma, 1 + 2u_\sigma)}. \tag{5.36}$$

To minimize the energy, we need $u_\Gamma = 1 \mod 2$, because otherwise there is an energy cost of order 1 from the sigma spins. This implies that both the symmetry and twist parameters of the tau spin are odd. From (5.31), all the energies of this kind degenerate. The minimal energy is thus achieved by choosing $u_\sigma = 0 \mod 2$ and $u_\Gamma = 1, 3 \mod 4$. In summary, the $\mathbb{Z}^\Gamma_4$ charge under $t_\Gamma = 1$ TBC is $u_\Gamma = 1, 3 \mod 4$. This is consistent with the discussion in the unperturbed case $h = 0$ in Section 5.4, where it has been shown that the ground state energy under $\mathbb{Z}^\Gamma_4$ twist has a non-trivial relative $\mathbb{Z}^\tau_2 \subset \mathbb{Z}^\Gamma_4$ charge.

3. $t_\Gamma = 2 \mod 4$: By (5.34), the energy of the ground state in the symmetry-twist sector is

$$E^\Gamma_{(u_\Gamma, 2)} = E^\sigma_{(u_\sigma, u_\Gamma)} + E^\tau_{(u_\Gamma + 2u_\sigma, 2 + 2u_\sigma)}. \tag{5.37}$$

By similar analysis, the minimal energy is achieved by choosing $u_\Gamma = 2 \mod 4$ and $u_\sigma = 1 \mod 2$. Hence the $\mathbb{Z}^\Gamma_4$ charge under $t_\Gamma = 2$ TBC is $u_\Gamma = 2 \mod 4$. This is again consistent with the discussion in the unperturbed case $h = 0$ in Section 5.4, where it has been shown that the ground state energy under $\mathbb{Z}^\tau_2$ twist has a relative $\mathbb{Z}^\Gamma_4$ charge $2 \mod 4$.

4. $t_\Gamma = 3 \mod 4$: By (5.34), the energy of the ground state in the symmetry-twist sector is

$$E^\Gamma_{(u_\Gamma, 3)} = E^\sigma_{(u_\sigma, u_\Gamma + 3)} + E^\tau_{(u_\Gamma + 2u_\sigma, 3 + 2u_\sigma)}. \tag{5.38}$$

By similar analysis, the minimal energy is achieved by choosing $u_\Gamma = 1, 3 \mod 4$ and $u_\sigma = 0 \mod 2$. Hence the $\mathbb{Z}^\Gamma_4$ charge under $t_\Gamma = 3$ TBC is $u_\Gamma = 1, 3 \mod 4$. This is again consistent with the discussion in the unperturbed case $h = 0$ in Section 5.4, where it has been shown that the ground state energy under odd $\mathbb{Z}^\Gamma_4$ twist has a relative $\mathbb{Z}^\tau_2$ charge 1 mod 2.

---

[7]The energy (5.29) is obtained in the continuum, and does not capture everything on the lattice. In particular, the absolute charge of the ground state under PBC from the continuum is always trivial, while on the lattice the ground state charge depends on $L \mod 8$. But we anticipate that the relative charge on the lattice and in the continuum match if we focus on $L = 0 \mod 8$.

In summary, we have reproduced the $\mathbb{Z}_4^\Gamma$ ground state charge under various $\mathbb{Z}_4^\Gamma$ twisted boundary conditions as in Section 5.4 which was also discussed in [4]. The novel feature here is that the method used here also works after turning on perturbation $h$, while the method in Section 5.4 does not apply for non-trivial perturbation even when $h$ is infinitesimally small. Since the non-trivial relative $\mathbb{Z}_4^\Gamma$ charge of the ground state under $\mathbb{Z}_4^\Gamma$ twisted boundary conditions is a key feature of the intrinsically gapless SPT, we take it as a strong support that the igSPT is robust under small perturbation of the kind (5.26).

## 5.6   Phase diagram

In the previous subsection, we only studied a small perturbation and determined the ground state symmetry properties under various twisted boundary conditions. The results suggested that the igSPT at $h = 0$ is robust for small $h$. In this subsection, we proceed to increase $h$ and study the phase diagram. We finally comment on the string order parameter.

**Phase diagram:**   We first determine the critical $h$ where the phase transitions of (5.33) take place. Because the location of the phase transition is not changed under discrete gauging, the value of $h_c$ can be inferred from the model (5.27) before the KT transformation. Since before the KT transformation the system is decoupled, i.e. $H_{\text{XX+SSB+pert}} = H_{\text{XX+pert}} + H_{\text{SSB+pert}}$, the phase transition can be determined separately for each part. We summarize the structure of two parts below.

1. For the XX model with a transverse field, as discussed in Appendix D, when $0 \leq h < 2$, the model is a free massless boson (or equivalently a free massless Dirac fermion). The Fermi velocity decreases as $h$ increases, and eventually goes to zero at $h = 2$. When $h > 2$, the transverse field takes place and the model is trivially gapped. The phase transition between a gapless free boson and a trivially gapped phase occurs at $h = 2$. This is the Lifshitz transition [51,52].

2. For the $\mathbb{Z}_2^\sigma$ Ising model with a transverse field, when $0 \leq h < 1$, the model is in a gapped $\mathbb{Z}_2^\sigma$ SSB phase. When $h > 1$, the model is in a trivially gapped phase. The transition occurs at $h = 1$, i.e. the Ising phase transition.

Combining the above, we identified two phase transitions of the igSPT with perturbation (5.33) at $h = 1$ and $h = 2$ respectively. The schematic phase diagram is shown in Figure 2.

To see the phases in each regime, we discuss the topological features in $0 \leq h < 1$, $1 < h < 2$ and $h > 2$ respectively.

1. $0 \leq h < 1$: The topological features have been determined in Section 5.5, which represents the intrinsically gapped SPT phase.

2. $1 < h < 2$: We proceed to discuss the topological features in the regime $1 < h < 2$, by repeating the same analysis in Section 5.5. Before the KT transformation, the energy

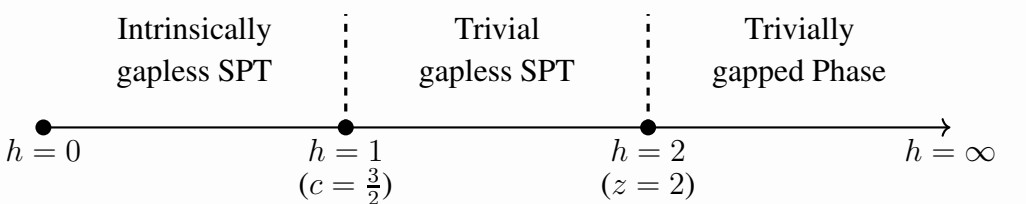

Figure 2: The phase diagram of (5.33).

hierarchy of the XX model with a transverse field is the same as (5.30), while that of the transverse field Ising model is modified to

$$E^\sigma_{(0,0)} \overset{e^{-L}}{<} E^\sigma_{(0,1)} \overset{\frac{1}{1}}{<} E^\sigma_{(1,0)} \overset{\frac{1}{L^2}}{<} E^\sigma_{(1,1)}. \tag{5.39}$$

This follows from the fact that the energy in $1 < h < 2$ is related to $\frac{1}{2} < h < 1$ via a Kramers-Wannier transformation, which exchanges the symmetry-twist sectors by $(u, t) \leftrightarrow (t, u)$. Then we combine (5.39), (5.30) and (5.34) to find the $\mathbb{Z}_4^\Gamma$ charge of the ground state for various $\mathbb{Z}_4^\Gamma$ twisted boundary conditions. From (5.34), we find that $u_\sigma = 0$ in order to avoid the order one energy cost (according to (5.39)),

$$E^\Gamma_{(u_\Gamma, t_\Gamma)} = E^\sigma_{(0, t_\Gamma + u_\Gamma)} + E^\tau_{(u_\Gamma, t_\Gamma)}. \tag{5.40}$$

The energy is then dominated by the $\tau$ spin. From (5.30), we further observe that for any $t_\Gamma$, the energy $E^\tau_{(u_\Gamma, t_\Gamma)}$ is minimized by $u_\Gamma = 0 \mod 4$. This concludes that the relative $\mathbb{Z}_4^\Gamma$ charge is always trivial. This implies that the Hamiltonian (5.33) with $1 < h < 2$ is a trivial gapless SPT!

3. $h > 2$: Only the transverse field term $-h \sum_{i=1}^L (\sigma_i^x + \tau_{i-\frac{1}{2}}^x)$ dominates in this regime, and hence the theory is in the trivially gapped phase.

We summarize the above discussion in the phase diagram in Figure 2. We note that the above phase diagram is also supported by exact numerical diagonalization studied in [4], for example, the transition at $h = 2$ is clearly indicated in Figure 5 of [4]. The jumps of the charges for $h < 2$ in the numerical results in [4] are essentially due to the subtleties of $L \neq 0 \mod 8$ and various finite size effects. The new KT transformation thus provides an analytic understanding of the stability of intrinsically gapless SPT under perturbation.

**String order parameter:** We finally comment on the string order parameter in the igSPT. Before the KT transformation, the decoupled Hamiltonian has the conventional correlation function:

$$\langle \sigma_i^z \sigma_j^z \rangle \sim \begin{cases} \mathcal{O}(1), & h < 1 \\ e^{-\xi_\sigma L}, & h > 1 \end{cases}, \qquad \langle \tau_{i-\frac{1}{2}}^z \tau_{j-\frac{1}{2}}^z \rangle \sim \begin{cases} \frac{1}{|i-j|^{2\Delta}}, & h < 2 \\ e^{-\xi_\tau L}, & h > 2 \end{cases} \tag{5.41}$$

where $\Delta$ and $\xi_\tau$ are the scaling dimension and the correlation length of $\tau^z$ respectively. Similar for $\xi_\sigma$.

After the KT transformation, the conventional correlation function becomes string order parameters:

$$\langle \sigma_i^z (\prod_{k=i}^{j-1} \tau_{k+\frac{1}{2}}^x) \sigma_j^z \rangle \sim \begin{cases} \mathcal{O}(1), & h < 1 \\ e^{-\xi_\sigma L}, & h > 1 \end{cases}, \qquad \langle \tau_{i-\frac{1}{2}}^z (\prod_{k=i}^{j-1} \sigma_k^x) \tau_{j-\frac{1}{2}}^z \rangle \sim \begin{cases} \frac{1}{|i-j|^{2\Delta}}, & h < 2 \\ e^{-\xi_\tau L}. & h > 2 \end{cases} \quad (5.42)$$

# 6  Purely gapless SPT and intrinsically purely gapless SPT from KT transformation

In Section 4 and Section 5, we discussed the construction of gapless SPT and intrinsically gapless SPT from the KT transformation, starting from the decoupled $\mathbb{Z}_2^\sigma$ SSB phase and a gapless theory (Ising CFT and XX model for gSPT and igSPT respectively). Since both cases contain a gapped sector ($\mathbb{Z}_2^\sigma$ SSB phase) before the KT transformation, the resulting gSPT and igSPT always contain a gapped sector. This is confirmed, for example by computing the energy splitting of the edge modes on an open chain [3, 6]. The gapped sectors can also be seen by the decorated domain wall construction explored in [2–4]. A natural question is can we construct a gapless SPT with no gapped sector?

In this section, we will construct both gapless SPT and intrinsically gapless SPT that do not contain gapped sectors, and follow [5] to denote them as underline{purely gapless SPT} (pgSPT) and underline{intrinsically purely gapless SPT} (ipgSPT) respectively. In particular, we begin with two decoupled gapless XXZ chains, and perform a KT transformation.

## 6.1  Constructing pgSPT and ipgSPT

Let us start with two decoupled XXZ chains, The Hamiltonian is

$$H_{\text{XXZ+XXZ}}^h = -\sum_{i=1}^{L} \left( \sigma_i^z \sigma_{i+1}^z + \sigma_i^y \sigma_{i+1}^y + h \sigma_i^x \sigma_{i+1}^x + \tau_{i-\frac{1}{2}}^z \tau_{i+\frac{1}{2}}^z + \tau_{i-\frac{1}{2}}^y \tau_{i+\frac{1}{2}}^y + h \tau_{i-\frac{1}{2}}^x \tau_{i+\frac{1}{2}}^x \right). \quad (6.1)$$

For pgSPT, we will focus on $\mathbb{Z}_2^\sigma \times \mathbb{Z}_2^\tau$, and for ipgSPT we will focus on the $\mathbb{Z}_2^\sigma \times \mathbb{Z}_4^\tau$ symmetry. Here $\mathbb{Z}_2^\sigma$ and $\mathbb{Z}_2^\tau$ are defined in (2.2) and $\mathbb{Z}_4^\tau$ is defined in (5.3). Below, we will mainly focus on the ipgSPT and the symmetry $\mathbb{Z}_2^\sigma \times \mathbb{Z}_4^\tau$. The pgSPT is obtained by the same theory and only changes the symmetry to be the normal subgroup $\mathbb{Z}_2^\sigma \times \mathbb{Z}_2^\tau$.

**Constructing ipgSPT with $\mathbb{Z}_4^\Gamma$ symmetry:**  Applying the KT transformation (associated with $\mathbb{Z}_2^\sigma \times \mathbb{Z}_2^\tau$ where the latter is the normal subgroup of $\mathbb{Z}_4^\tau$), the decoupled Hamiltonian becomes

$$H_{\text{ipgSPTpert}}^h = -\sum_{i=1}^{L} \left( \sigma_i^z \tau_{i+\frac{1}{2}}^x \sigma_{i+1}^z + \sigma_i^y \tau_{i+\frac{1}{2}}^x \sigma_{i+1}^y + h \sigma_i^x \sigma_{i+1}^x + \tau_{i-\frac{1}{2}}^z \sigma_i^x \tau_{i+\frac{1}{2}}^z + \tau_{i-\frac{1}{2}}^y \sigma_i^x \tau_{i+\frac{1}{2}}^y + h \tau_{i-\frac{1}{2}}^x \tau_{i+\frac{1}{2}}^x \right).$$

$$(6.2)$$

Since the KT transformation commutes with both $\sigma_i^x$ and $\tau_{i-\frac{1}{2}}^x$, the resulting Hamiltonian (6.2) also has $\mathbb{Z}_2^\sigma \times \mathbb{Z}_4^\tau$ symmetry, generated by $U_\sigma$ and $V_\tau$. We will again focus on the $\mathbb{Z}_4^\Gamma$ subgroup, generated by $U_\sigma V_\tau$.

The Hamiltonian (6.2) depends on a parameter $h$, which can be taken to be either positive or negative. Below, we will study the $\mathbb{Z}_4^\Gamma$ symmetry properties of the ground state of (6.2) under various $\mathbb{Z}_4^\Gamma$ twisted boundary conditions to determine the regime of $h$ where the Hamiltonian is topologically non-trivial, hence is ipgSPT. It will become clear in the following subsections that the ipgSPT corresponds to $0 < h < 1$.

Before determining the topological properties of the phases, it is useful to see where the phase transitions can take place, which can be inferred from the decoupled theory (6.1) before the KT transformation.

From Appendix D, we know that the phase transition occurs at $h = -1$ and $h = 1$. Later we will also see that there is a more subtle phase transition at $h = 0$.

**pgSPT with symmetry $\mathbb{Z}_2^\sigma \times \mathbb{Z}_2^\tau$:** The Hamiltonian of pgSPT is given by the same Hamiltonian (6.2). The only difference is the choice of symmetry, which is taken to be $\mathbb{Z}_2^\sigma \times \mathbb{Z}_2^\tau$. Here $\mathbb{Z}_2^\tau$ is the normal subgroup of $\mathbb{Z}_4^\tau$. It turns out that the non-trivial pgSPT also corresponds to the parameter regime $0 < h < 1$.

## 6.2 Field theory description

**Field theory of ipgSPT:** Before studying the ground state property, we first comment on the field theory description of the lattice ipgSPT Hamiltonian (6.2). As reviewed in Appendix D, the field theory of the XXZ model is a free boson [50]. Hence we begin with the partition function

$$Z_{\text{boson}}^h[A_\tau] Z_{\text{boson}}^h[A_\sigma] \tag{6.3}$$

where $A_\tau$ is a $\mathbb{Z}_4$ cocycle, and $A_\sigma$ is a $\mathbb{Z}_2$ cocycle. More explicitly the two decoupled partition functions are

$$
\begin{aligned}
Z_{\text{boson}}^h[A_\tau] &:= \int \mathcal{D}\theta_\tau \exp\left(i \int_{X_2} \frac{1}{2\pi K_h}(D_{A_\tau}\theta_\tau)^2\right), & D_{A_\tau}\theta_\tau &= d\theta_\tau - i\frac{\pi}{2}A_\tau\theta_\tau \\
Z_{\text{boson}}^h[A_\sigma] &:= \int \mathcal{D}\theta_\sigma \exp\left(i \int_{X_2} \frac{1}{2\pi K_h}(D_{A_\sigma}\theta_\sigma)^2\right), & D_{A_\sigma}\theta_\sigma &= d\theta_\sigma - i\pi A_\sigma\theta_\sigma.
\end{aligned}
\tag{6.4}
$$

We then perform a $STS$ transformation, changing the partition function to

$$Z_{\text{ipgSPTpert}}^h[A_\sigma, A_\tau] = \sum_{\substack{a_\sigma = 0,1 \\ a_\tau = A_\tau \mod 2}} Z_{\text{boson}}^h[a_\sigma] Z_{\text{boson}}^h[a_\tau] e^{\frac{i\pi}{2}\int_{X_2}(A_\tau - a_\tau)(A_\sigma + a_\sigma)} \tag{6.5}$$

which is a gauged version of two decoupled free bosons. By further restricting the $\mathbb{Z}_2^\sigma \times \mathbb{Z}_4^\tau$ symmetry to $\mathbb{Z}_4^\Gamma$, whose background fields are identified as $A_\tau = A_\Gamma \mod 4$, and $A_\sigma = A_\Gamma$

mod 2, we have

$$Z^h_{\text{ipgSPTpert}}[A_\Gamma] = \sum_{\substack{a_\sigma=0,1 \\ a_\tau=A_\Gamma \mod 2}} Z^h_{\text{boson}}[a_\sigma] Z^h_{\text{boson}}[a_\tau] e^{\frac{i\pi}{2} \int_{X_2} (A_\Gamma - a_\tau)(A_\Gamma + a_\sigma)}. \tag{6.6}$$

It is also useful to see why the construction in [3, 4] does not apply here. Let us decompose $A_\Gamma$ as $A_\Gamma = 2B + A$, with $\delta B = A^2 \mod 2$, then the partition function simplifies to

$$Z^h_{\text{ipgSPTpert}}[A_\Gamma] = \sum_{\substack{a,b=0,1 \\ \delta b = A^2 \mod 2}} Z^h_{\text{boson}}[a] Z^h_{\text{boson}}[2b + A] e^{i\pi \int_{X_2} ab + aB + bA + AB}. \tag{6.7}$$

Note that the partition function can not be factorized into $Z_{\text{low}}[A] Z_{\text{gapped}}[A, B]$ where the low energy sector only depends on the quotient $\mathbb{Z}_2$. Hence the entire $\mathbb{Z}_4$ symmetry couples to the low energy degrees of freedom and there is no obvious gapped sector.

**Field theory of pgSPT:** The field theory of pgSPT can be obtained by starting from $Z^h_{\text{boson}}[A_\tau] Z^h_{\text{boson}}[A_\sigma]$ where both $A_\tau$ and $A_\sigma$ are $\mathbb{Z}_2$ valued background fields, and perform a KT transformation. The resulting partition function is

$$Z^h_{\text{pgSPTpert}}[A_\sigma, A_\tau] = \sum_{a_\sigma, a_\tau = 0,1} Z^h_{\text{boson}}[a_\sigma] Z^h_{\text{boson}}[a_\tau] e^{i\pi \int_{X_2} (A_\tau + a_\tau)(A_\sigma + a_\sigma)}. \tag{6.8}$$

Again there is also no decoupled gapped sector because both $A_\sigma$ and $A_\tau$ couple to the gapless sectors via magnetic coupling.

## 6.3 Topological features of pgSPT and ipgSPT from KT transformation and phase diagram

In this subsection, we proceed to study the topological features of (6.2). Since there is no term in (6.2) which commutes with the rest of the terms, the method from [4] (which were also reviewed in Section 4.3 and Section 5.4) does not work. Hence we directly apply the KT transformation to study the topological features. We will be mainly discussing the topological features and the phase diagram of ipgSPT as a function of $h$. We will also briefly comment on the phase diagram of pgSPT.

We begin by studying the topological features of two copies of XXZ chain with anisotropy parameter $h$. Note that the global symmetry of the sigma spin is $\mathbb{Z}_2^\sigma$, by repeating the discussion in Appendix D, we find the ground state energy in the $\mathbb{Z}_2^\sigma$ symmetry-twist sector $E^\sigma_{(u_\sigma, t_\sigma)}$ is

$$E^\sigma_{(u_\sigma, t_\sigma)}(h) = \frac{2\pi}{L} \left[ \frac{1}{4K_h} [u_\sigma]_2^2 + \frac{K_h}{4} [t_\sigma]_2^2 \right]. \tag{6.9}$$

Note that $[u_\sigma]_2$ is $u_\sigma$ modulo 2, and similar for $[t_\sigma]_2$. For the $\tau$ spin, the global symmetry is $\mathbb{Z}_4^\tau$, and the ground state energy in the $\mathbb{Z}_4^\tau$ symmetry-twist sector is given by (D.22),

$$E^\sigma_{(v_\tau, r_\tau)}(h) = \frac{2\pi}{L} \left[ \frac{1}{4K_h} (\min([v_\tau]_4, 4 - [v_\tau]_4))^2 + \frac{K_h}{16} (\min([r_\tau]_4, 4 - [r_\tau]_4))^2 \right]. \tag{6.10}$$

Here $K_h$ is related to $h$ as follows,

$$-1 < h < 0 \Leftrightarrow \frac{1}{2} < K_h < 1, \qquad 0 < h < 1 \Leftrightarrow K_h > 1. \qquad (6.11)$$

Note that in the regime $h < -1$ and $h > 1$, the XXZ model is gapped and spontaneously breaks the $\mathbb{Z}_2^y$ symmetry which is generated by the $\prod_i \sigma_i^y$ [53, 54]. Hence the cosine terms in the Sine-Gordon model can not be ignored as they are relevant operators.

We proceed to the system (6.2) after the KT transformation, and consider the ground state in each $\mathbb{Z}_4^\Gamma$ symmetry-twist sector $E_{(u_\Gamma, t_\Gamma)}^\Gamma(h)$. The energy is the same as (5.34). Substituting (6.9) and (6.10) into (5.34), we obtain

$$
\begin{aligned}
E_{(u_\Gamma, t_\Gamma)}^\Gamma(h) =& E_{(u_\sigma, u_\Gamma + t_\Gamma)}^\sigma(h) + E_{(u_\Gamma + 2u_\sigma, t_\Gamma + 2u_\sigma)}^\tau(h) \\
=& \frac{L}{2\pi} \Bigg[ \frac{1}{4K_h}[u_\sigma]_2^2 + \frac{K_h}{4}[u_\Gamma + t_\Gamma]_2^2 + \frac{1}{4K_h}(\min([u_\Gamma + 2u_\sigma]_4, 4 - [u_\Gamma + 2u_\sigma]_4))^2 \\
& + \frac{K_h}{16}(\min([t_\Gamma + 2u_\sigma]_4, 4 - [t_\Gamma + 2u_\sigma]_4))^2 \Bigg].
\end{aligned}
$$

$$(6.12)$$

The structure of the minimal energy spectrum is as follows.

1. When $t_\Gamma = 0$, the lowest energy is achieved by $u_\Gamma = 0$ and $u_\sigma = 0$.

2. When $t_\Gamma = 1$, the lowest energy is achieved by $(u_\sigma, u_\Gamma) = (0, 0)$ if $K_h < 1$, and $(u_\sigma, u_\Gamma) = (0, 1)$ or $(0, 3)$ if $K_h > 1$.

3. When $t_\Gamma = 2$, the lowest energy is $(u_\sigma, u_\Gamma) = (0, 0)$ if $K_h < 1$, and $(u_\sigma, u_\Gamma) = (1, 2)$ if $K_h > 1$.

4. When $t_\Gamma = 3$, the lowest energy is achieved by $(u_\sigma, u_\Gamma) = (0, 0)$ if $K_h < 1$, and $(u_\sigma, u_\Gamma) = (0, 1)$ or $(0, 3)$ if $K_h > 1$.

Combining the correspondence between $h$ and $K_h$, we find that when $-1 < h < 0$, the ground state of the Hamiltonian (6.2) always has trivial $\mathbb{Z}_4^\Gamma$ charge $u_\Gamma = 0$ under any twisted boundary condition, hence it is in the trivial gapless phase. When $0 < h < 1$, the ground state of the Hamiltonian (6.2) has non-trivial relative $\mathbb{Z}_4^\Gamma$ charge under $\mathbb{Z}_4^\Gamma$ twisted boundary conditions, hence it is in the topologically non-trivial ipgSPT phase. We summarize the phase diagram in Figure 3.

We summarize the properties of the various phase transitions.

1. $h = -1$: The phase transition at $h = -1$ can be described by the Sine-Gordon model at $K_h = \frac{1}{2}$. This phase transition is described by the SU(2)$_1$ WZW CFT, where the cosine term $\cos(2\varphi)$ becomes marginal, between the irrelevant $h > -1$ ($K_h > \frac{1}{2}$) to relevant $h < -1$ ($K_h < \frac{1}{2}$).

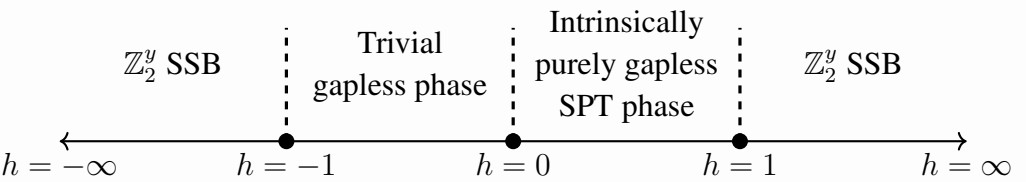

Figure 3: The phase diagram of ipgSPT.

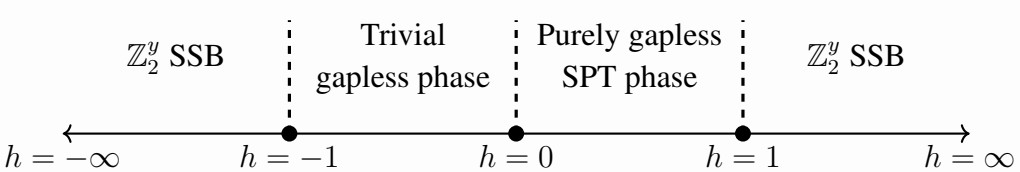

Figure 4: The phase diagram of pgSPT.

2. $h = 1$: The phase transition at $h = 1$ corresponds to $K_h \to \infty$, and is not described within the free boson description. But from the lattice model, the transition can be understood intuitively as the XX term dominates over the other terms and triggers the system to a gapped phase where $\mathbb{Z}_2^y$ is SSB.

3. $h = 0$: Unlike the transitions at $h = \pm 1$, this transition is not directly implied from the two decoupled XXZ chains before the KT transformation because both $K_h > 1$ and $K_h < 1$ are described by the free bosons with trivial topological features. However, after the KT transformation, the topological feature becomes non-trivial for $K_h > 1$ while still remains trivial for $K_h < 1$. This implies that there is a underline{topological phase transition} at $K_h = 1$, i.e. $h = 0$.

**String order parameter:** We comment on the string order parameter in the gapless region. Before the KT transformation, each decoupled Hamiltonian has two quasi long-range orders when $-1 < h < 1$. One is the correlation function of Pauli Z operators and the other is the disorder order parameter of Pauli X operators:

$$\langle \tau_{i-\frac{1}{2}}^z \tau_{j-\frac{1}{2}}^z \rangle = \langle \sigma_i^z \sigma_j^z \rangle \sim \frac{1}{|i-j|^{\frac{1}{2K_h}}}, \qquad \langle \prod_{k=i}^{j} \tau_{k-\frac{1}{2}}^x \rangle = \langle \prod_{k=i}^{j} \sigma_k^x \rangle \sim \frac{1}{|i-j|^{\frac{K_h}{2}}}. \qquad (6.13)$$

Thus, after the KT transformation, the first two quasi long-range orders become string order parameters

$$\langle \tau_{i-\frac{1}{2}}^z (\prod_{k=i}^{j-1} \sigma_k^x) \tau_{j-\frac{1}{2}}^z \rangle = \langle \sigma_i^z (\prod_{k=i}^{j-1} \tau_{k+\frac{1}{2}}^x) \sigma_j^z \rangle \sim \frac{1}{|i-j|^{\frac{1}{2K_h}}}, \qquad \langle \prod_{k=i}^{j} \tau_{k-\frac{1}{2}}^x \rangle = \langle \prod_{k=i}^{j} \sigma_k^x \rangle \sim \frac{1}{|i-j|^{\frac{K_h}{2}}}. \qquad (6.14)$$

Note that string order parameters carry a nontrivial symmetry charge on the end while the disorder charge does not. When $1 < K_h$ ($0 < h < 1$), the string order parameters decay lower than disorder parameters, while when $\frac{1}{2} < K_h < 1$ ($-1 < h < 0$), the string order parameters decay faster than disorder parameters. Here we remark that this result is consistent with ground state charge under twisted boundary conditions due to the state operator correspondence [3].

We also comment on the boundary degeneracy under OBC. Since the pgSPT is obtained from two decoupled XXZ chains, the low-energy spectrum of an XXZ chain under OBC in the gapless region, similar to that under PBC, scales as $\mathcal{O}(1/L)$. Based on the mapping by the KT transformation, we expect that the spectrum of pgSPT constructed in this Section also scales as $\mathcal{O}(1/L)$, under either PBC or OBC. This curious observation should be contrasted from the $\mathcal{O}(1/L^{14})$ finite size gap decaying behavior under OBC of the pgSPT with a different anti-unitary symmetry—$\mathbb{Z}_2 \times \mathbb{Z}_2^T$ [6], which clearly differs from $\mathcal{O}(1/L)$ under PBC. Nevertheless, various other signatures including non-trivial symmetry charge under TBC as well as the symmetry charge of the string order parameter suggest non-trivial SPT features, hence we still consider our model as a different type of pgSPT. We will leave a more elaborated discussion on pgSPT to the future.

**Phase diagram of pgSPT:** We finally comment on the pgSPT. The discussions are mostly in parallel to the ipgSPT, and we will not give the details here. The topological features of the purely gapless SPTs are such that the $\mathbb{Z}_2^\sigma$ ($\mathbb{Z}_2^\tau$) symmetry charge of ground state under $\mathbb{Z}_2^\tau$ ($\mathbb{Z}_2^\sigma$) twisted boundary condition is non-trivial. It is also interesting to note that the decaying behavior of the string order parameter does not change qualitatively for the trivial gapless phase and the pgSPT since they both decay polynomially, hence the phase transition is subtle as we have seen above from a separate perspective. The final phase diagram is shown in Figure 4.

# Acknowledgement

We would like to thank Philip Boyle Smith, Weiguang Cao, Yoshiki Fukusumi and Hong Yang for useful discussions. L.L. is supported by Global Science Graduate Course (GSGC) program at the University of Tokyo. Y.Z. is partially supported by WPI Initiative, MEXT, Japan at IPMU, the University of Tokyo. This work was supported in part by MEXT/JSPS KAKENHI Grants No. JP17H06462 and No. JP19H01808, and by JST CREST Grant No. JPMJCR19T2.

# A   Energy Spectrum of Transverse Field Ising model under $\mathbb{Z}_2$ TBC

The Hamiltonian of transverse field Ising model is

$$H_{\text{Ising}} = -\sum_{i=1}^{L}(\sigma_i^z \sigma_{i+1}^z + h\sigma_i^x) \tag{A.1}$$

with $\mathbb{Z}_2$ twisted boundary condition: $\sigma^z_{i+L} = -\sigma^z_i, \sigma^x_{i+L} = \sigma^x_i$.

We apply the Jordan-Wigner (JW) transformation which maps the spin operator to the fermion operator

$$\sigma^x_i = (-1)^{n_i} = 1 - 2f^\dagger_i f_i, \quad \sigma^z_i = \prod_{j=1}^{i-1}(-1)^{n_j}(f^\dagger_i + f_i) \tag{A.2}$$

where $n_i := f^\dagger_i f_i$ is the fermion density operator. Note that when $i = 1$, we simply have $\sigma^z_1 = f^\dagger_1 + f_1$.

Applying the JW transformation to the lsing model, we can rewrite (A.1) in terms of the fermions,

$$H_{\text{lsing}} = -hL - \sum_{i=1}^{L}\left(-2hf^\dagger_i f_i + (f^\dagger_i - f_i)(f^\dagger_{i+1} + f_{i+1})\right) \tag{A.3}$$

with boundary condition

$$f_{i+L} = (-1)^F f_i, \quad F = \sum_{j=1}^{L} n_j. \tag{A.4}$$

After Fourier transformation and Bogoliubov transformation, this Hamiltonian is diagonal

$$H_{\text{lsing}} = \sum_k \omega_k \left(c^\dagger_k c_k - \frac{1}{2}\right) \tag{A.5}$$

where $\omega_k = 2\sqrt{1 - 2h\cos k + h^2}$. Moreover, the fermion parity after transformation is given by

$$(-1)^{\sum_{0<k<\pi} c^\dagger_k c_k + c^\dagger_{-k} c_{-k}} = (-1)^{\sum_{0<k<\pi} f^\dagger_k f_k + f^\dagger_{-k} f_{-k}},$$
$$(-1)^{c^\dagger_0 c_0} = (-1)^{f^\dagger_0 f_0} sign(h - 1), \quad (-1)^{c^\dagger_\pi c_\pi} = (-1)^{f^\dagger_\pi f_\pi} \tag{A.6}$$

When $h = 1$, there is a zero mode with $k = 0$ and whether it is realizable depends on the boundary condition. Moreover, the fermion parity of the $k = 0$ modes after Bogoliubov transformation changes depends on $h$. We discuss them separately.

## Energy spectrum of SSB phase ($0 \leq h < 1$)

If $(-1)^F = 1$, the fermion chain has PBC. This means $k = \frac{2\pi j}{L}$ where $j = 0, \cdots, L-1$. Therefore, we have $k = 0$ mode when $j = 0$. As the total fermion parity after Bogoliubov transformation changes, the ground states are: $c^\dagger_\pi |\text{VAC}\rangle_{\text{PBC}}$. The ground state energy is

$$E^{\text{PBC}}_{\text{GS}} = -\frac{1}{2}\sum_{k=\frac{2\pi j}{L}} \omega_k + 2|1 - h|. \tag{A.7}$$

If $(-1)^F = -1$, the fermion chain has anti-periodic boundary condition (ABC) where $k = \frac{(2j+1)\pi}{L}$ and there is no $k = 0$ mode. The ground states are $c^\dagger_{-\frac{\pi}{L}} |\text{VAC}\rangle_{\text{ABC}}$ and $c^\dagger_{\frac{\pi}{L}} |\text{VAC}\rangle_{\text{ABC}}$ with ground state energy:

$$E^{\text{ABC}}_{\text{GS}} = -\frac{1}{2} \sum_{k=\frac{(2j+1)\pi}{L}} \omega_k + 2\sqrt{1 + h^2 - 2h \cos\frac{\pi}{L}}. \tag{A.8}$$

When $L$ is large, the first term of $E^{\text{ABC}}_{\text{GS}}$ and $E^{\text{PBC}}_{\text{GS}}$ has a gap of order $e^{-L}$ and the second term has a gap of order $\frac{1}{L^2}$. Thus we have $E^{\text{ABC}}_{\text{GS}} > E^{\text{PBC}}_{\text{GS}}$ and the gap is of order $\frac{1}{L^2}$.

In summary, the ground state of the transverse field Ising model when $h < 1$ is unique. Since $\prod^L_{j=1} \sigma^x_j = (-1)^F$, the ground state is always in the even sector while the first excited states are in the odd sector and the finite size gap is $\frac{1}{L^2}$.

## Energy spectrum of trivial phase ($h > 1$)

If $(-1)^F = 1$, the fermion chain has PBC, then $k = \frac{2\pi j}{L}$ where $j = 0, \cdots, L-1$. When $j = 0$, we have $k = 0$ mode. Since the total fermion parity after Bogoliubov transformation is invariant, the ground state is $|\text{VAC}\rangle_{\text{PBC}}$ with ground state energy:

$$E^{\text{PBC}}_{\text{GS}} = -\frac{1}{2} \sum_{k=\frac{2\pi j}{L}} \omega_k. \tag{A.9}$$

If $(-1)^F = -1$, the fermion chain has ABC where $k = \frac{(2j+1)\pi}{L}$. And the total fermion parity after the Bogoliubov transformation is also invariant. Since $(-1)^F = -1$, the ground states are $c_{-\frac{\pi}{L}} |\text{VAC}\rangle_{\text{ABC}}$ and $c_{\frac{\pi}{L}} |\text{VAC}\rangle_{\text{ABC}}$ with ground state energy:

$$E^{\text{ABC}}_{\text{GS}} = -\frac{1}{2} \sum_{k=\frac{(2j+1)\pi}{L}} \omega_k + 2\sqrt{1 + h^2 - 2h \cos\frac{\pi}{L}}. \tag{A.10}$$

Therefore, we obtain that $E^{\text{PBC}}_{\text{GS}} < E^{\text{ABC}}_{\text{GS}}$ and the gap is finite in thermodynamic limit. Then the ground state of the transverse field Ising model when $h > 1$ is unique and gapped. Moreover, the ground state is always in the even sector while the first excited states are in the odd sector.

## Energy spectrum of critical point ($h = 1$)

If $(-1)^F = 1$, the fermion chain has PBC, then $k = \frac{2\pi j}{L}$ where $j = 0, \cdots, L-1$. We also have $k = 0$ mode when $j = 0$. Since the total fermion parity after Bogoliubov transformation is invariant, the ground state is $|\text{VAC}\rangle_{\text{PBC}}$ with ground state energy:

$$E^{\text{PBC}}_{\text{GS}} = -\frac{1}{2} \sum_{k=\frac{2\pi j}{L}} \omega_k = -2 \sum_{k=\frac{2\pi j}{L}} |\cos(\frac{k}{2})| = -2\cot(\frac{\pi}{2L}). \tag{A.11}$$

If $(-1)^F = -1$, the fermion chain has ABC where $k = \frac{(2j+1)\pi}{L}$. And the total fermion parity after the Bogoliubov transformation is also invariant. Since $(-1)^F = -1$, the ground states are $c_{\frac{(2L-1)\pi}{2L}} |\text{VAC}\rangle_{\text{ABC}}$ and $c_{\frac{(2L+1)\pi}{2L}} |\text{VAC}\rangle_{\text{ABC}}$ with ground state energy:

$$E_{\text{GS}}^{\text{ABC}} = -\frac{1}{2} \sum_{k = \frac{(2j+1)\pi}{L}} \omega_k + 4 \sin \frac{\pi}{2L} = -\frac{2}{\sin(\frac{\pi}{2L})} + 4 \sin \frac{\pi}{2L}. \tag{A.12}$$

Therefore, we obtain that $E_{\text{GS}}^{\text{PBC}} < E_{\text{GS}}^{\text{ABC}}$ and the finite size gap is of order $\frac{1}{L}$. Then the ground state of the lsing model at the critical point under TBC is always in the $\mathbb{Z}_2$ even sector while the first excited states are in the $\mathbb{Z}_2$ odd sector and the finite size gap is of order $\frac{1}{L}$.

# B  Stability of edge mode of gSPT and igSPT phase on the OBC

In this appendix, we will discuss the stability of the finite size gap of gapless SPT and intrinsically gapless SPT on the OBC under the symmetric boundary perturbation. We aim to prove that there are at least two nearly degenerate ground states with finite size gap of order $e^{-L}$ and this (exponential) degeneracy is stable under any symmetric boundary perturbation.

Let's first consider the gapless SPT phase in section 4, and focus on the two decoupled systems before KT transformation. One is a $\mathbb{Z}_2$ SSB model with $\tau$ spin and the other is the critical transverse field Ising model lsing model with $\sigma$ spin. Thus the low energy states are:

$$|\text{even}_\tau\rangle \otimes |\psi_j^\sigma\rangle, \quad |\text{odd}_\tau\rangle \otimes |\psi_j^\sigma\rangle \tag{B.1}$$

where $|\text{even}_\tau\rangle$ and $|\text{odd}_\tau\rangle$ are nearly degenerate ground states of $\tau$ spins with even and odd $\mathbb{Z}_2^\tau$ charge and the finite size gap $E_{\text{odd}}^\tau - E_{\text{even}}^\tau$ is of order $e^{-L}$. $\psi_j^\sigma$ is the energy eigenstate of the gapless $\sigma$ spin model which is labeled by the positive integer $j$. In this basis, the low energy effective Hamiltonian is blocked diagonal:

$$H_{\text{SSB+gapless}}^{\text{low}} = \begin{pmatrix} E_{\text{even}}^\tau + H_{\text{gapless}}^\sigma & 0 \\ 0 & E_{\text{odd}}^\tau + H_{\text{gapless}}^\sigma \end{pmatrix}. \tag{B.2}$$

Moreover, any other excited state of $\tau$ spin has a finite gap $\Delta_{\text{SSB}}$ in the thermodynamic limit. It is obvious that this system has at least two ground states with exponential energy splitting. Due to the unitarity of KT transformation on an open chain, this implies the exponential finite size gap between edge modes.

Now let us add a $\mathbb{Z}_2^\sigma \times \mathbb{Z}_2^\tau$ symmetric perturbation $hV$ in the gapless SPT Hamiltonian. Since symmetry operators are both invariant under KT transformation, the perturbation in the decoupled system before KT transformation is also $\mathbb{Z}_2^\sigma \times \mathbb{Z}_2^\tau$ symmetric. As the KT transformation is unitary on the open chain, the energy spectrum before and after the KT transformation are the same and we will focus on the energy spectrum of the decoupled system with the perturbation above.

At first, we can decompose $hV$ as follows:

$$hV = h \sum_{i=1}^N V_\sigma^i V_\tau^i \tag{B.3}$$

where $N$ can be polynomial in $L$. $V_\sigma^i$ and $V_\tau^i$ only act on finite range of the $\sigma$ and $\tau$ spin respectively. When $h \ll \Delta_{\text{SSB}}$, we consider the first perturbation theory, namely we only need to consider how this perturbation acts on the low energy states. Since $hV_\tau^i$ is symmetric, it is diagonal for $|\text{even}_\tau\rangle$ and $|\text{odd}_\tau\rangle$ :

$$hV_\tau^i = \begin{pmatrix} ha_{\text{even}}^i & 0 \\ 0 & ha_{\text{odd}}^i \end{pmatrix}, \tag{B.4}$$

where $a_{\text{even}}^i = \langle\text{even}_\tau| V_\tau^i |\text{even}_\tau\rangle$ and $a_{\text{odd}}^i = \langle\text{odd}_\tau| V_\tau^i |\text{odd}_\tau\rangle$ is finite independent of system size. Thus the low energy effective Hamiltonian with this perturbation is

$$H_{\text{SSB+gapless}} + hV = \begin{pmatrix} E_{\text{even}}^\tau + H_{\text{gapless}}^\sigma + h\sum_{i=1}^N a_{\text{even}}^i V_\sigma^i & 0 \\ 0 & E_{\text{odd}}^\tau + H_{\text{gapless}}^\sigma + h\sum_{i=1}^N a_{\text{odd}}^i V_\sigma^i \end{pmatrix}. \tag{B.5}$$

Moreover, as $h \ll \Delta_{\text{SSB}}$, the $\tau$ spin chain after adding $hV_\tau^i$ is still in the SSB phase and the finite size gap $ha_{\text{even}}^i - ha_{\text{odd}}^i$ is of order $e^{-L}$.

In the next step, we denote the ground states of $H_{\text{gapless}}^\sigma + h\sum_{i=1}^N a_{\text{even}}^i V_\sigma^i$ and $H_{\text{gapless}}^\sigma + h\sum_{i=1}^N a_{\text{odd}}^i V_\sigma^i$ as $|\text{GS}_\sigma\rangle$ and $|\text{GS}_\sigma'\rangle$ with energy $E_1$ and $E_1'$ respectively. Without loss of generality, we can assume $E_1 \leq E_1'$. Then we notice that

$$\langle\text{GS}_\sigma| H_{\text{gapless}}^\sigma + h\sum_{i=1}^N a_{\text{odd}}^i V_\sigma^i - H_{\text{gapless}}^\sigma - h\sum_{i=1}^N a_{\text{even}}^i V_\sigma^i |\text{GS}_\sigma\rangle \propto \langle\text{GS}_\sigma| e^{-L}\sum_{i=1}^N V_\sigma^i |\text{GS}_\sigma\rangle \propto e^{-L}. \tag{B.6}$$

Note that the sum over $i$ does not change exponential decaying behaviour of finite size gap.

On the other hand, we also have

$$\langle\text{GS}_\sigma| H_{\text{gapless}}^\sigma + h\sum_{i=1}^N a_{\text{odd}}^i V_\sigma^i - H_{\text{gapless}}^\sigma - h\sum_{i=1}^N a_{\text{even}}^i V_\sigma^i |\text{GS}_\sigma\rangle \tag{B.7}$$

$$= \quad \langle\text{GS}_\sigma| H_{\text{gapless}}^\sigma + h\sum_{i=1}^N a_{\text{odd}}^i V_\sigma^i |\text{GS}_\sigma\rangle - E_1 \tag{B.8}$$

$$\geq \quad (E_1' - E_1) \geq 0 \tag{B.9}$$

Thus, we can obtain that $E_1' - E_1$ is of order $e^{-L}$. Since $E_{\text{odd}}^\tau - E_{\text{even}}^\tau$ is also of order $e^{-L}$, the total finite size gap between $|\text{even}_\tau\rangle \otimes |\text{GS}_\sigma\rangle$ and $|\text{odd}_\tau\rangle \otimes |\text{GS}_\sigma'\rangle$ is of order $e^{-L}$, which finishes our proof.

For the intrinsically gapless SPT phase in section 5, if we add a $\mathbb{Z}_4^\Gamma$ symmetric perturbation, this perturbation may be mapped to a nonlocal perturbation under KT transformation. For example, $\sigma_L^z(\tau_{L-\frac{1}{2}}^z\tau_{L+\frac{1}{2}}^z - \tau_{L-\frac{1}{2}}^y\tau_{L+\frac{1}{2}}^y) \to (\prod_{j<L} \tau_{j+\frac{1}{2}}^x)\sigma_L^z\sigma_L^x(\tau_{L-\frac{1}{2}}^z\tau_{L+\frac{1}{2}}^z - \tau_{L-\frac{1}{2}}^y\tau_{L+\frac{1}{2}}^y)$. Thus, we will consider $\mathbb{Z}_2^\sigma \times \mathbb{Z}_4^\tau$ symmetric perturbations. which is still local and $\mathbb{Z}_2^\sigma \times \mathbb{Z}_4^\tau$ symmetric under the KT transformation. Then the proof of the stability of edge modes is similar to that of the gapless SPT phase above and we don't repeat it here.

# C  Mapping $\mathbb{Z}_4^\tau \times \mathbb{Z}_2^\sigma$ symmetry-twist sectors under the KT transformation

In this appendix, we derive the mapping between $\mathbb{Z}_4^\tau \times \mathbb{Z}_2^\sigma$ symmetry-twist sectors under KT transformation directly from the partition function. We start with a theory $\mathcal{X}$ with an anomaly-free $\mathbb{Z}_4^\tau \times \mathbb{Z}_2^\sigma$ symmetry, and denote their background fields as $A_\tau$ and $A_\sigma$. The partition function is $Z_\mathcal{X}[A_\sigma, A_\tau]$. The KT transformation is the twisted gauging of the $\mathbb{Z}_2^\sigma$ and $\mathbb{Z}_2^\tau$ normal subgroup of $\mathbb{Z}_4^\tau$. Since only the normal subgroup of $\mathbb{Z}_4^\tau$ participates in gauging, it is useful to first decompose the background field into $A_\tau = 2B_\tau + C_\tau$, where both $B_\tau$ and $C_\tau$ are $\mathbb{Z}_2$ valued 1-cochains, satisfying the condition

$$\delta A_\sigma = 0 \mod 2, \qquad \delta C_\tau = 0 \mod 2, \qquad \delta B_\tau = C_\tau^2 \mod 2. \tag{C.1}$$

Under KT transformation, the partition function of the resulting theory is

$$\sum_{\widetilde{a}_\sigma, \widetilde{b}_\tau, a_\sigma, b_\tau} Z_\mathcal{X}[a_\sigma, 2b_\tau + C_\tau] e^{i\pi \int_{X_2} a_\sigma \widetilde{b}_\tau + b_\tau \widetilde{a}_\sigma + \widetilde{b}_\tau \widetilde{a}_\sigma + \widetilde{a}_\sigma B_\tau + \widetilde{b}_\tau A_\sigma}$$

$$= \sum_{a_\sigma, b_\tau} Z_\mathcal{X}[a_\sigma, 2b_\tau + C_\tau] e^{i\pi \int_{X_2} (b_\tau + B_\tau)(a_\sigma + A_\sigma)}. \tag{C.2}$$

In the second line, we summed over $\widetilde{a}_\sigma$ and $\widetilde{b}_\tau$. What symmetry does the resulting theory have? To see this, we check whether the resulting partition function (C.2) depends on the 3d bulk. The dynamical part $Z_\mathcal{X}$ is clearly independent of the 3d bulk since $\mathcal{X}$ is anomaly free. By promoting the remaining part to the 3d integral using using derivative, and applying the bundle constraints $\delta a_\sigma = 0 \mod 2, \delta C_\tau = 0 \mod 2, \delta b_\tau = C_\tau^2 \mod 2$ which follow from (C.1), the 3d dependence is

$$e^{i\pi \int_{X_3} (C_\tau^2 + \delta B_\tau)(a_\sigma + A_\sigma) + (b_\tau + B_\tau)\delta A_\sigma}. \tag{C.3}$$

For the resulting theory to be an absolute theory, we need to demand that all terms involving dynamical fields to vanish. This in particular requires

$$\delta B_\tau = C_\tau^2 \mod 2, \qquad \delta A_\sigma = 0 \mod 2. \tag{C.4}$$

Once these conditions are imposed, all the 3d dependence is trivialized. One can then introduce again a $\mathbb{Z}_4^\tau$ connection, such that This shows that after KT transformation, the theory still has a $\mathbb{Z}_2^\sigma \times \mathbb{Z}_4^\tau$ symmetry, and they are also anomaly free.

Denoting the partition function after the KT transformation as $Z_{\widehat{\mathcal{X}}}[\widehat{A}_\sigma, \widehat{A}_\tau]$, it is related to the partition function of $\mathcal{X}$ via

$$Z_{\widehat{\mathcal{X}}}[\widehat{A}_\sigma, \widehat{A}_\tau] = \sum_{\substack{a_\sigma = 0,1, \\ a_\tau = \widehat{A}_\tau \mod 2}} Z_\mathcal{X}[a_\sigma, a_\tau] e^{\frac{i\pi}{2} \int_{X_2} (\widehat{A}_\tau - a_\tau)(\widehat{A}_\sigma + a_\sigma)}. \tag{C.5}$$

In terms of holonomies around the time and space directions, the partition function can be rewritten as

$$Z_{\mathcal{X}}[(W_t^\sigma, W_x^\sigma), (W_t^\tau, W_x^\tau)] := Z_{\mathcal{X}}[A_\sigma, A_\tau] \tag{C.6}$$

where $W_t^\sigma, W_x^\sigma \in \mathbb{Z}_2$ and $W_t^\tau, W_x^\tau \in \mathbb{Z}_4$.

As discussed in Section 5.2, the partition function can also be labeled by $[(u_\sigma, t_\sigma), (v_\tau, r_\tau)]$. Hence we denote the partition function as $Z_{\mathcal{X}}^{((u_\sigma, t_\sigma), (v_\tau, r_\tau))}$. Its relation with $Z_{\mathcal{X}}[(W_t^\sigma, W_x^\sigma), (W_t^\tau, W_x^\tau)]$ is

$$Z_{\mathcal{X}}^{((u_\sigma, t_\sigma), (v_\tau, r_\tau))} = \frac{1}{8} \sum_{\substack{w_t^\sigma = 0,1 \\ w_t^\tau = 0,1,2,3}} Z_{\mathcal{X}}[(w_t^\sigma, t_\sigma), (w_t^\tau, r_\tau)] e^{i\pi w_t^\sigma u_\sigma + \frac{i\pi}{2} w_t^\tau v_\tau} \tag{C.7}$$

and the converse relation is

$$Z_{\mathcal{X}}[(W_t^\sigma, W_x^\sigma), (W_t^\tau, W_x^\tau)] = \sum_{\substack{u_\sigma = 0,1 \\ v_\tau = 0,1,2,3}} Z_{\mathcal{X}}^{((u_\sigma, w_x^\sigma), (v_\tau, w_x^\tau))} e^{i\pi w_t^\sigma u_\sigma - \frac{i\pi}{2} w_t^\tau v_\tau}. \tag{C.8}$$

By combining (C.5), (C.6), (C.7) and (C.8), we find the relation between $Z_{\widehat{\mathcal{X}}}^{((\widehat{u}_\sigma, \widehat{t}_\sigma), (\widehat{v}_\tau, \widehat{r}_\tau))}$ and $Z_{\mathcal{X}}^{((u_\sigma, t_\sigma), (v_\tau, r_\tau))}$,

$$Z_{\widehat{\mathcal{X}}}^{((\widehat{u}_\sigma, \widehat{t}_\sigma), (\widehat{v}_\tau, \widehat{r}_\tau))} = Z_{\mathcal{X}}^{((\widehat{u}_\sigma, \widehat{t}_\sigma + \widehat{v}_\tau), (\widehat{v}_\tau, \widehat{r}_\tau + 2\widehat{u}_\sigma))} \equiv Z_{\mathcal{X}}^{((u_\sigma, t_\sigma), (v_\tau, r_\tau))}. \tag{C.9}$$

Hence the symmetry and twist sectors are related as

$$[(\widehat{u}_\sigma, \widehat{t}_\sigma), (\widehat{v}_\tau, \widehat{r}_\tau)] = [(u_\sigma, t_\sigma - v_\tau), (v_\tau, r_\tau + 2u_\sigma)]. \tag{C.10}$$

# D   XX model with a transverse field, XXZ model, and free boson

In this appendix, we discuss the XX model with two types of perturbations, and discuss the symmetry properties of their ground states under various twisted boundary conditions. These results will be useful in Section 5 and 6.

## D.1   XX model and free boson

We begin with the XX model, whose Hamiltonian is

$$H_{\text{XX}} = -\sum_{i=1}^{L} \sigma_i^z \sigma_{i+1}^z + \sigma_i^y \sigma_{i+1}^y. \tag{D.1}$$

We introduce the ladder operators as $\sigma_i^\pm = (\sigma_i^z \pm i\sigma_i^y)/2$, the Hamiltonian can be rewritten as

$$H_{\text{XX}} = -\sum_{i=1}^{L} 2\sigma_i^+ \sigma_{i+1}^- + 2\sigma_i^- \sigma_{i+1}^+. \tag{D.2}$$

It has a $U(1)$ global symmetry, but we will only focus on its $\mathbb{Z}_4$ subgroup. The symmetry operator is

$$U = \prod_{i=1}^{L} e^{\frac{i\pi}{4}(1-\sigma_i^x)} \tag{D.3}$$

whose eigenvalue is $e^{\frac{i\pi}{2}u}$, where $u = 0, 1, 2, 3$. The symmetry operator $U$ acts on the ladder operators as $U^\dagger \sigma_i^\pm U = e^{\pm\frac{i\pi}{2}}\sigma_i^\pm$. Thus the twisted boundary condition is specified by

$$\sigma_{i+L}^\pm = e^{\pm\frac{i\pi}{2}t}\sigma_i^\pm, \qquad t = 0, 1, 2, 3. \tag{D.4}$$

We would like to consider the continuous limit of this theory. It is well-known that via Jordan-Wigner (JW) transformation, the XX model is equivalent to a free fermion model. To see this, we consider the JW transformation[8]

$$\sigma_i^x = 1 - 2f_i^\dagger f_i, \qquad \sigma_i^+ = \prod_{j=1}^{i-1}(-1)^{f_j^\dagger f_j}f_j, \qquad \sigma_i^- = \prod_{j=1}^{i-1}(-1)^{f_j^\dagger f_j}f_j^\dagger. \tag{D.5}$$

The Hamiltonian then becomes a free fermion

$$H_{\text{fer}} = -\sum_{i=1}^{L} 2f_i^\dagger f_{i+1} + 2f_{i+1}^\dagger f_i. \tag{D.6}$$

After taking the Fourier transformation, the fermion Hamiltonian is diagonalized, which takes the form

$$H_{\text{fer}} = -\sum_k 4\cos\left(\frac{2\pi}{L}k\right)f_k^\dagger f_k \tag{D.7}$$

where $k \simeq k+L$, and the fractional value of $k$ depends on the boundary conditions of the fermion, which will not be important for our purpose.

The energy spectrum is $E_k = -4\cos\left(\frac{2\pi}{L}k\right)$, which intersects the Fermi surface at two Fermi points $k \simeq \pm\frac{L}{4}$. The ground state is given by filling all electrons in the band within $k \in [-\frac{L}{4}, \frac{L}{4}]$, and the low energy excitations are all localized around the two Fermi points. After linearizing the energy spectrum around the Fermi points, we find a right moving Weyl fermion at $k = \frac{L}{4}$ and a left moving Weyl fermion at $k = -\frac{L}{4}$. The two Weyl fermions compose into a Dirac fermion. At the field theory level, the bosonization of a free Dirac fermion gives rise to a $c = 1$ free boson. The Lagrangian of the free boson is

$$\mathcal{L} = \frac{1}{2\pi}(\partial_\mu\theta)^2 = \frac{1}{2\pi}\left[(\partial_t\theta)^2 - (\partial_x\theta)^2\right]. \tag{D.8}$$

Upon canonical quantization, we have $P_\theta = \frac{\partial_t\theta}{\pi}$, and $[\theta, P_\theta] = i$. It is customary to introduce the dual variable $\varphi$ such that $P_\theta = \frac{\partial_x\varphi}{2\pi}$, in terms of which the Hamiltonian becomes

$$H_{\text{boson}} = \int dx(P_\theta\dot\theta - \mathcal{L}) = \frac{1}{2\pi}\int dx\left[\frac{1}{4}(\partial_x\varphi)^2 + (\partial_x\theta)^2\right]. \tag{D.9}$$

---

[8]Note that for convenience we changed the sign in the first relation compared to (A.2).

In terms of the lattice variables, we have $\sigma_i^{\pm} \simeq e^{\pm i\theta}$, and the $\mathbb{Z}_4$ symmetry operator is

$$U = e^{-\frac{i}{4}\int dx \partial_x \varphi}. \tag{D.10}$$

Indeed, using the BCH formula, $U^{\dagger} e^{i\theta} U = e^{i\theta + \frac{i\pi}{2}}$, which is consistent with the commutation relation on the lattice $U^{\dagger} \sigma_i^+ U = e^{\frac{\pi i}{2}} \sigma_i^+$.

The fields $\varphi(x,t)$ and $\theta(x,t)$ are subjected to the twisted boundary condition,

$$\varphi(x+L,t) = \varphi(x,t) + 2\pi m, \qquad \theta(x+L,t) = \theta(x,t) + 2\pi n. \tag{D.11}$$

After mode expansion, the fields are decomposed into zero modes and oscillator modes. Hence we have

$$\varphi(x,t) \simeq 2\pi m \frac{x}{L} + \cdots, \qquad \theta(x,t) \simeq 2\pi n \frac{x}{L} + \cdots \tag{D.12}$$

where $m, n$ are constrained by the twisted boundary conditions for $\varphi$ and $\theta$ respectively, which will consequently be determined by the charge and symmetry twists on the lattice. The ground state energy only receives a contribution from the zero modes, which gives $\frac{2\pi}{L}[\frac{1}{4}m^2 + n^2]$.

To see how $m, n$ are constrained, we first notice that the eigenvalue of $U = e^{-\frac{i}{4}\int dx \partial_x \varphi}$ is $e^{\frac{i\pi}{2}u}$. Substituting (D.12) into $U$, we find

$$m = -u + 4m' \tag{D.13}$$

where $m'$ is an integer. On the other hand, the twisted boundary condition $\sigma_{i+L}^+ = e^{\frac{\pi i}{2}t} \sigma_i^+$ on the lattice implies the twisted boundary boundary condition of $\theta(x,t)$ in the continuum, hence

$$n = \frac{1}{4}t + n' \tag{D.14}$$

where $n'$ is an integer. This implies that the zero mode energy is

$$E_{(u,t)}^{m',n'} = \frac{2\pi}{L} \left[ \frac{1}{4}(4m' - u)^2 + (n' + \frac{t}{4})^2 \right]. \tag{D.15}$$

Let us denote the ground state in the symmetry-twist sector as $E_{(u,t)}$, obtained by minimizing (D.15) overall $m', n'$, we have

$$E_{(u,t)} = \frac{2\pi}{L} \left[ \frac{1}{4}(\min([u]_4, 4 - [u]_4))^2 + \frac{1}{16}(\min([t]_4, 4 - [t]_4))^2 \right]. \tag{D.16}$$

where $[u]_4$ stands for the $u$ mod 4, and similar for $[t]_4$. In particular, for any $t$, the minimal ground state always carries trivial $\mathbb{Z}_4$ charge, i.e. $u = 0$.

## D.2   Adding a transverse field

We further perturb the XX model (D.1) by a transverse field $-h \sum_{i=1}^{L} \sigma_i^x$, such that the total Hamiltonian is[9]

$$H_{\text{XXpert}} = -\sum_{i=1}^{L} \sigma_i^z \sigma_{i+1}^z + \sigma_i^y \sigma_{i+1}^y + h\sigma_i^x \tag{D.17}$$

---

[9]Naively, one may attempt to simply add $h\partial_x \varphi$ to the Lagrangian (D.8), but this does not work. The reason is that turning on $h$ changes the location of the Fermi point, while $h\sigma_i^x \sim \frac{h}{\pi}\partial_x \varphi$ holds only near the original Fermi point $h \sim \frac{L}{4}$, hence is expanding around the wrong vacuum.

To see the property of the energy spectrum, it is again useful to perform a JW transformation, which shows that the energy is $E_k = -4\cos\left(\frac{2\pi}{L}k\right) + 2h$. The transverse field plays a role of shifting energy of the entire band by $2h$. As long as $h < 2$, the band still intersects $E = 0$ with two Fermi points, hence the system is still gapless. The effective degrees of freedom are still two Weyl fermions at the two Fermi points, but the only difference is that they are at closer momenta, and the Fermi velocities are reduced (determined by $\frac{dE_k}{dk}$). We then re-bosonize at the field theory level, and obtain a free boson, whose Lagrangian is

$$\mathcal{L} = \frac{1}{2\pi}(\partial_\mu \theta)^2 = \frac{1}{2\pi}\left[\frac{1}{v_h}(\partial_t \theta)^2 - v_h(\partial_x \theta)^2\right], \qquad v_h = \sqrt{1 - \frac{h^2}{4}}. \tag{D.18}$$

Indeed, when $h = 0$, $v_0 = 1$, which is consistent with (D.8). When $h \to 2$, the Fermi velocity reduces to zero, corresponding to the point where the band of the fermion is tangential to the $E_k = 0$ axis, i.e. the two Fermi-points merge. When $h > 2$, the Fermi velocity is imaginary showing that the description (D.18) breaks down. Indeed, whe $h > 2$, the term $-h\sigma_i^x$ dominates, driving the system to a trivially gapped phase. Because the transition at $h = 2$ is associated with a quadratic band, the transition has dynamical exponent $z = 2$.

We proceed to discuss the ground state properties. In the free boson representation, the Hamiltonian is

$$H_{\text{boson}} = \frac{v_h}{2\pi}\int dx \left[\frac{1}{4}(\partial_x \varphi)^2 + (\partial_x \theta)^2\right]. \tag{D.19}$$

The energy is exactly the same as (D.15) except for an overall normalization by the Fermi velocity. Thus the symmetry charges of the ground state under various twisted boundary conditions are the same as the unperturbed case $h = 0$, as long as $h < 2$.

## D.3  XXZ model and free boson

We can alternatively perturb the XX model (D.1) by $-h\sum_{i=1}^L \sigma_i^x \sigma_{i+1}^x$. The total Hamiltonian is

$$H_{\text{XXZ}} = -\sum_{i=1}^L \sigma_i^z \sigma_{i+1}^z + \sigma_i^y \sigma_{i+1}^y + h\sigma_i^x \sigma_{i+1}^x. \tag{D.20}$$

When $h = 1$ or $-1$, the Hamiltonian is the ferromagnetic/anti-ferromagnetic Heisenberg chain. When $h = 0$, the Hamiltonian reduces to the XX model. For other values of $h$, the Hamiltonian is the XXZ model.

The continuum field theory for the XXZ model is well known. When $-1 < h < 1$, it is the gapless Luttinger liquid whose Hamiltonian is [10]

$$H_{\text{LL}} = \frac{1}{2\pi}\int dx \left[\frac{1}{4K_h}(\partial_x \varphi)^2 + K_h(\partial_x \theta)^2\right]. \tag{D.21}$$

---

[10]More precisely, the low energy effective theory also contains a $\cos(2\phi)$ term. Such a term is irrelevant for $K_h > \frac{1}{2}$ hence we ignore it in the low energy. However this term is relevant for $K_h < \frac{1}{2}$, which gaps out the Hamiltonian to obtain a SSB phase.

Here the Luttinger parameter is $K_h = \frac{\pi}{2\arccos h}$ [55]. Therefore when $h < 0$, the system is described by $K_h < 1$ free boson, while when $h > 0$ the system is described by $K_h > 1$ free boson.

When $h > 1$ or $h < -1$, the $\sigma^x \sigma^x$ term in (D.20) dominates [53, 54], and there are two nearly degenerate ground states with an exponentially decaying gap. Note that the two degenerate ground states spontaneously break the $\mathbb{Z}_2$ generated by $\prod_i \sigma_i^z$ or $\prod_i \sigma_i^y$.

The energy coming from the zero mode is almost the same as (D.16), but with the Luttinger parameter adjusted,

$$E_{(u,t)} = \frac{2\pi}{L} \left[ \frac{1}{4K_h} (\min([u]_4, 4 - [u]_4))^2 + \frac{K_h}{16} (\min([t]_4, 4 - [t]_4))^2 \right]. \tag{D.22}$$

In particular, for any $t$, the minimal ground state always carries trivial $\mathbb{Z}_4$ charge, i.e. $u = 0$.

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
