# Peer review of "Intrinsically/Purely Gapless-SPT from Non-Invertible Duality Transformations"

_SciPost Physics_

## Round 1 · Referee Report · Anonymous (Referee 1) · 2023-11-24

Report

Although the classification of gapped symmetry-protected topological (SPT) phases is by now well-established, the study of SPT effects in gapless systems is still an open avenue of research. Several kinds of gapless SPT orders have been introduced over the recent years, including gapless SPT, intrinsically gapless SPT, and purely gapless SPT.

The authors provide a unifying perspective on these effects, mostly focusing on spin-1/2 chains with Z_2 x Z_2 symmetry. This unified description relies on the Kennedy-Tasaki transformation, which provides a powerful approach to study these questions. The wide range of results (Eqs 1.1 to 1.9) they manage to obtain using this technique is very impressive and provides a clear physical picture for the rich physics associating gaplessness and SPT order in such systems. They also provide the first example (to the best of my knowledge) of an intrinsically purely gapless SPT.

The paper is well-written and very pedagogical. I thus recommend publication, provided the authors address my comment below:

• As far as I can tell, an explanation of the notation of gauge degrees of freedom is not given. For example, above Eq 3.4, Z[A_sigma,A_tau] is used but A_sigma,A_tau are not defined. For the sake of the paper being self-contained, could the authors explain more clearly their notation for the gauging procedure, and for formulas like Eq 3.5? They also use an integration over “X_2” which is not explicitly defined I believe? I think they probably use notation they introduced in an earlier paper, but it would be good to redefine it here.

  • validity: -
  • significance: -
  • originality: -
  • clarity: -
  • formatting: -
  • grammar: -

Author:  Li Linhao  on 2025-04-13  [id 5360]

(in reply to Report 1 on 2023-11-24)

We thank the referees for the illuminating comments. Please find our response below.

Reply to Referre 1 We would like to thank the referee for the helpful comments. We have added an explanation of the notation of gauge degrees of freedom, and added a footnote which cites some references for discrete gauge fields in the updated version.

Reply to referee 2. We would like to thank the referee for the helpful suggestions and comments. We have add a decription of general structures of various types of gapless SPTs in the introductions. The following is our response to each comment.

  1. At first, we want to clarify that the Trivial to SPT transition is not obtained by stacking an Ising^2 CFT by a Z2 x Z2 SPT. The latter corresponds to the phase transition between Z2 x Z2 SPT and Z2 x Z2 SSB phase. One lattice model realizing Z2 x Z2 SPT to trivial phase transition is

    $$H=- \sum_{j} (\sigma^z_{j-1} \tau^x_{j-\frac12} \sigma^z_{j} + \tau^z_{j-\frac12} \sigma^x_j \tau^z_{j+\frac12} + \sigma^x_j + \tau^x_{j-\frac12})$$
    . The Z2 x Z2 symmetry is generated by $U_\sigma=\prod_j \sigma^x_j$ and $U_\tau= \prod_j \tau^x_{j-\frac12}$. Under the twisted boundary condition of the first Z2, there are two ground states, carrying the opposite charge under the second Z2. Hence does not satisfy the topological feature discussed in this paper. (We require the gSPT ground state have a definite symmetry charge under twisted boundary condition.)

  2. We have revised the paper to add the coefficient before SSB phases and transformations.

  3. Indeed, there can be an anomalous Z2 symmetry in 1+1d bosonic systems. The anomaly of Z_low in 5.19 comes from that \int_{X_2}bA=\int{M_3}A^3 where X_2 is the boundary of M_3 and we use the fact \delta b=A^2. This topological term is anomaly flow action of anomalous Z2 symmetry which lives on the bounary of 2+1d Z2 SPT (Levin-Gu model on the lattice).

  4. The ground state degeneracy of intrinsically gapless SPT (igSPT) model Eq.(5.5) is 2 when system size L is 2 mod 4, while it is 1 in other case. Indeed, the low energy spectrum of igSPT is the same as that of boundary of Levin-Gu model (H_LG=-\sum_i(X_i-Z_i-1X_iZ_i+1), including ground state degenracy. For the boundary Levin-Gu model, we can understand its ground state degeneracy from ZY chain H_ZY=-(Z_iZ_i+1+Y_iY_i+1) as they are related by the KW transformation. When size is odd, H_ZY has two ground states according to the anticommutation between symmetry operators \prod_i X_i and \prod_i Z_i. We can choose two ground states with \prod_i X_i=\pm 1. When size is even, the ground state is unique with zero magentization \sum_i X_i=0 and the first excited states are two degenerate with \sum_i X_i=\pm 2. As \prod_n X_n=i^L\exp(i\pi sum_n X_n/2), the ground state has symmetry eigenvalue i^L and the first excited states have symmetry eigenvalue -(i^L). Under the KW transformation, only states with even eigenvalue. Thus the boundary Levin-Gu model has two ground state when L is 1 mod 4 and unique ground state in other cases.

  5. We have edited the typos.

---

## Round 1 · Referee Report · Anonymous (Referee 2) · 2023-12-11

Strengths

This is a very interesting and timely research topic.

Weaknesses

This paper describes a lot of detailed examples with little clarification of the underlying conceptual ideas.

Report

The authors address a very interesting and timely subject of topological aspects of gapless phases, which have gotten a fair bit of attention in the recent past.

The paper makes copious use of the Kennedy-Tasaki transformation, which essentially implements a combination of gauging the Z2 x Z2 symmetry with pasting an SPT. Using this transformation and some knowledge of the c=1/2 Ising and c=1 free boson transitions, the authors are able to study various topological aspects in a doubled spin-1/2 chain with Z2 x Z2 symmetry.

Despite the interesting analysis, it is the referee’s feeling that the underlying structure could be described a bit more in detail. As it is written, the authors go quickly into examples and sometimes the reader is left missing the underlying conceptual view.

A suggestion— perhaps the authors would add a section clarifying the general structure of various types of gapless SPTs. This is partially done in the introduction and particularly table 1 and eq 1.2-1.9, but more clean definitions of these would be welcome.

Requested changes

Some comments/questions:

1) Perhaps this is a naive question. Why is the Triv to SPT transition in 1.5 not a gapless SPT. It is obtained by stacking an Ising^2 CFT by a Z2 x Z2 SPT. Is it because the entire symmetry acts on the gapless sector. If so, it would be very nice if the authors could provide a clean definition of gapless SPTs.

2) In many places in the paper it is written Z_SSB =delta(A). Is a factor of 2 (related to the ground state degeneracy) missing?

3) Around 5.19, what is the nature of this anomaly. It is known that there are no anomalies for Z2 symmetry in bosonic systems. Perhaps these can be cancelled by some local counter terms. could the authors please comment on this ?

4) Is there a symmetry based explanation of the size dependent GSD in the intrinsically gapless SPT?

5) The section heading of Sec. 2.2 has a typo.

  • validity: high
  • significance: good
  • originality: good
  • clarity: ok
  • formatting: reasonable
  • grammar: reasonable

Author:  Li Linhao  on 2025-04-13  [id 5365]

(in reply to Report 2 on 2023-12-11)

We would like to thank the referee for the helpful suggestions and comments. We have add a decription of general structures of various types of gapless SPTs in the introductions. The following is our response to each comment.

  1. At first, we want to clarify that the Trivial to SPT transition is not obtained by stacking an Ising^2 CFT by a Z2 x Z2 SPT. The latter corresponds to the phase transition between Z2 x Z2 SPT and Z2 x Z2 SSB phase. One lattice model realizing Z2 x Z2 SPT to trivial phase transition is

    $$H=- \sum_{j} (\sigma^z_{j-1} \tau^x_{j-\frac12} \sigma^z_{j} + \tau^z_{j-\frac12} \sigma^x_j \tau^z_{j+\frac12} + \sigma^x_j + \tau^x_{j-\frac12})$$
    . The Z2 x Z2 symmetry is generated by $U_\sigma=\prod_j \sigma^x_j$ and $U_\tau= \prod_j \tau^x_{j-\frac12}$. Under the twisted boundary condition of the first Z2, there are two ground states, carrying the opposite charge under the second Z2. Hence does not satisfy the topological feature discussed in this paper. (We require the gSPT ground state have a definite symmetry charge under twisted boundary condition.)

  2. We have revised the paper to add the coefficient before SSB phases and transformations.

  3. Indeed, there can be an anomalous $Z_2$ symmetry in 1+1d bosonic systems. The anomaly of Z_low in 5.19 comes from that $\int_{X_2}bA=\int{M_3}A^3$ where $X_2$ is the boundary of $M_3$ and we use the fact $\delta b=A^2$. This topological term is anomaly flow action of anomalous $Z_2$ symmetry which lives on the bounary of 2+1d $Z_2$ SPT (Levin-Gu model on the lattice).

  4. The ground state degeneracy of intrinsically gapless SPT (igSPT) model Eq.(5.5) is 2 when system size L is 2 mod 4, while it is 1 in other case. Indeed, the low energy spectrum of igSPT is the same as that of boundary of Levin-Gu model, i.e., $H_{LG}=-\sum_i(X_i-Z_{i-1}X_iZ_{i+1}$), including ground state degenracy. For the boundary Levin-Gu model, we can understand its ground state degeneracy from ZY chain $H_{ZY}=-\sum(Z_iZ_{i+1}+Y_iY_{i+1})$ as they are related by the KW transformation. When size is odd, H_ZY has two ground states according to the anticommutation between symmetry operators $\prod_i X_i$ and $\prod_i Z_i$. We can choose two ground states with $\prod_i X_i=\pm 1$. When size is even, the ground state is unique with zero magentization $\sum_i X_i=0$ and the first excited states are two degenerate with $\sum_i X_i=\pm 2$. As $\prod_n X_n=i^L\exp(i\pi \sum_n X_n/2)$, the ground state has symmetry eigenvalue $i^L$ and the first excited states have symmetry eigenvalue $-(i^L)$. Under the KW transformation, only states with even eigenvalue survive. Thus the boundary Levin-Gu model has two ground state when L is 2 mod 4 and unique ground state in other cases.

  5. We have edited the typos.

---

## Editorial Decision

resubmitted